# `GDeR`: Safeguarding Efficiency, Balancing, and Robustness via Prototypical Graph Pruning

**Guibin Zhang**[*1,2], **Haonan Dong**[*1], **Yuchen Zhang**[2], **Zhixun Li**[3], **Dingshuo Chen**[4],
**Kai Wang**[5], **Tianlong Chen**[6], **Yuxuan Liang**[7], **Dawei Cheng**[†1,2], **Kun Wang**[†8]

[1]Tongji Univerity, [2]Shanghai AI Laboratory, [3]CUHK, [4]UCAS,
[5]NUS, [6]UNC-Chapel Hill, [7]HKUST (Guangzhou) [8]NTU
[*] Equal Contribution, [†] Corresponding author
dcheng@tongji.edu.cn, wk520529wjh@gmail.com

## Abstract

Training high-quality deep models necessitates vast amounts of data, resulting in overwhelming computational and memory demands. Recently, data pruning, distillation, and coreset selection have been developed to streamline data volume by *retaining*, *synthesizing*, or *selecting* a small yet informative subset from the full set. Among these methods, data pruning incurs the least additional training cost and offers the most practical acceleration benefits. However, it is the most vulnerable, often suffering significant performance degradation with imbalanced or biased data schema, thus raising concerns about its accuracy and reliability in on-device deployment. Therefore, there is a looming need for a new data pruning paradigm that maintains the efficiency of previous practices while ensuring balance and robustness. Unlike the fields of computer vision and natural language processing, where mature solutions have been developed to address these issues, graph neural networks (GNNs) continue to struggle with increasingly large-scale, imbalanced, and noisy datasets, lacking a unified dataset pruning solution. To achieve this, we introduce a novel dynamic soft-pruning method, `GDeR`, designed to update the training "basket" during the process using trainable prototypes. `GDeR` first constructs a well-modeled graph embedding hypersphere and then samples *representative, balanced, and unbiased subsets* from this embedding space, which achieves the goal we called `Graph Training Debugging`. Extensive experiments on five datasets across three GNN backbones, demonstrate that `GDeR` (I) achieves or surpasses the performance of the full dataset with $30\% \sim 50\%$ fewer training samples, (II) attains up to a $2.81\times$ lossless training speedup, and (III) outperforms state-of-the-art pruning methods in imbalanced training and noisy training scenarios by $0.3\% \sim 4.3\%$ and $3.6\% \sim 7.8\%$, respectively. The source code is available at https://github.com/ins1stenc3/GDeR.

## 1 Introduction

Data-centric AI, though continuously providing high-quality data for upcoming artificial general intelligence [1, 2, 3, 4], presents a significant hurdle for their on-device deployment during training and inference phases [5, 6, 7, 8]. To democratize existing state-of-the-art methods [9, 10, 9, 11, 12, 13, 14], considerable efforts are directed toward identifying unbiased and core data within training datasets and conducting troubleshooting to deepen our solid understanding of the intrinsic property of data shcema [15, 16, 17, 18]. To date, *data pruning* [19, 20, 21, 22, 23], *distillation* [24, 25, 26, 27, 28, 29, 30] and *coreset selection* [31, 32, 33, 16] aim to retain, synthesize or choose a small but informative dataset from original full set. While the sample size undergoes significant reshaping and reduction, methods like dataset distillation inevitably lead to additional training costs [23, 34, 35]. As a hardware-friendly candidate and accelerator for training and inference, data pruning serves as a promising candidate by mitigating the high computational burden.

38th Conference on Neural Information Processing Systems (NeurIPS 2024).

Recent advancements, however, have demonstrated that the efficiency of data pruning may come at a cost—utilizing only a portion of the data can potentially render the model more vulnerable to imbalance or malicious perturbation attacks [37], which are commonly seen in real-world applications. As illustrated in Figure 1, we evaluate the performance of the current state-of-the-art data pruning paradigm, InfoBatch [23], with a pruning ratio of 50%, on MUTAG [38]+GCN [39] under both imbalance and biased scenarios. It can be observed that: ❶ InfoBatch exacerbates the imbalance of training samples during the training process; ❷ InfoBatch efficiently saves training costs in noise-free scenarios, even surpassing the original full dataset performance by 0.7%. However, it encounters significant performance degradation (5.7% ∼ 6.4% ↓) in biased scenarios.

In this paper, we primarily focus on *graph-level data pruning*, aiming to enhance the model's

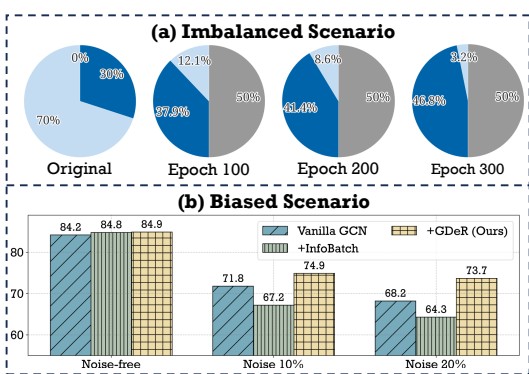

Figure 1: (**a**) We report the label distribution of the training set retained by InfoBatch at pruning ratios of 50% in the {0, 100, 200, 300}-th epochs. The gray, light blue and dark blue represent pruned, minority, and majority samples, respectively. (**b**) Performance comparison between InfoBatch and our GDeR when introducing outliers (following [36]) into {0%, 10%, 20%} of the training set.

robustness to data imbalance and noise while maintaining the efficiency inherited from traditional data pruning practices. This is because, unlike in computer vision (CV) and natural language processing (NLP) domains, where separate solutions already exist for addressing these issues [37, 23], graph learning models continue to grapple with increasingly large-scale, imbalanced, and biased datasets [40, 36, 41]. To this end, we first introduce a novel direction in the realm of graph training, termed Graph Training Debugging (GTD), to (dynamically) identify *representative, robust, and unbiased subsets* for accelerating the training process without compromising performance.

We achieve GTD goal by proposing a novel dynamical soft-pruning method, Graph De-Redundancy (GDeR), in which specifically designed to work efficiently and accurately on various GNN architectures. Concretely, GDeR draws inspiration from prototype learning [42, 43] practices, projecting training graph samples onto a hyperspherical embedding space. It utilizes a set of trainable prototypes to regularize the graph embedding distribution, essentially encouraging both inter-class separateness and intra-class compactness. Furthermore, on this well-regularized hypersphere, GDeR generates a sampling distribution that encourages the sampling of under-learned graphs, while excluding those with high outlier risk and belonging to majority clusters. Given a training budget (*i.e.*, pruning ratio), GDeR dynamically maintains a sub-dataset at each epoch, efficiently combating the negative impact of imbalanced and noisy data on the model, simultaneously accelerating training significantly.

**Broader Impact.** In this paper, we present a novel training philosophy GDeR to achieve our defined GTD goal. GDeR dynamically prunes irrelevant graph samples, providing a more comprehensive insight and achieving a triple-win of *efficiency*, *balancing*, and *robustness*. This approach can contribute to a wide range of graph-related applications, accelerating model training while demonstrating great potential in scenarios such as adversarial attacks [44, 45, 45], imbalanced graph classification [40, 46], and unsupervised pre-training [47, 48, 49]. We believe GDeR can serve as a benchmark for future research in this area, attracting significant attention and inspiring further exploration into understanding sparsity in other domains such as LLMs.

**Experimental Observation** We validate the GDeR through a comprehensive series of graph-level tasks, across five datasets and three GNN backbones, showcasing that GDeR can: ❶ achieve lossless training performances with 30% ∼ 50% fewer training samples, ❷ achieve a 2.0× lossless speedup on OGBG-MOLHIV, and a 2.81× lossless speedup on pre-training ZINC. ❸ mitigate imbalance issues by achieving a 0.3 ∼ 4.3% ↑ in F1-macro on MUTAG and DHFR datasets, ❹ effectively help outlier-attacked GNNs improve accuracy by 3.5% ∼ 10.2% through data pruning.

**Limitations & Future Insight.** GDeR, as a plug-in to graph training, not only improves efficiency but also ensures robustness and balance throughout the training. However, the applicability of its principles in fields such as CV remains unexplored, limiting the generalizability of data debugging. This represents a direction for future development in our work.

## 2 Technical Background

**Notations**   Consider an undirected graph $\mathcal{G} = (\mathcal{V}, \mathcal{E})$, where $\mathcal{V}$ represents the node set and $\mathcal{E}$ signifies the edges. The feature matrix for the graph is designated as $\mathbf{X} \in \mathbb{R}^{|\mathcal{V}| \times F}$. Each node $v_i \in \mathcal{V}$ is associated with a feature vector of $F$ dimensions. The adjacency matrix $\mathbf{A} \in \{0, 1\}^{N \times N}$ represents the connectivity between nodes, where $\mathbf{A}[i, j] = 1$ suggests the presence of an edge $e_{ij} \in \mathcal{E}$, and 0 indicates no edge. In graph-level training tasks, specifically for graph classification, given a set of $N$ graphs $\{\mathcal{G}\} = \{\mathcal{G}_1, \mathcal{G}_2, \ldots, \mathcal{G}_N\}$, where each graph $\mathcal{G}_i = (\mathcal{V}^i, \mathcal{E}^i)$ is as defined above, and their corresponding labels $\mathbf{Y} \in \mathbb{R}^{N \times C}$ with $C$ being the total number of classes, we aim to learn graph representations $\mathbf{H} \in \mathbb{R}^{N \times d'}$ with $\mathbf{H}[i, :]$ for each $\mathcal{G}_i \in \mathcal{G}$ that effectively predict $\mathbf{Y}_i$.

**Graph Neural Networks (GNNs).**   GNNs [50, 51] have become pivotal for learning graph representations, achieving benchmark performances in various graph tasks at *node-level* [52], *edge-level* [53], and *graph-level* [54]. The success of GNN mainly stems from message-passing mechanism:

$$\mathbf{h}_i^{(l)} = \texttt{COMB}\left(\mathbf{h}_i^{(l-1)}, \texttt{AGGR}\{\mathbf{h}_j^{(l-1)} : v_j \in \mathcal{N}(v_i)\}\right), \ 0 \leq l \leq L. \tag{1}$$

Here, $L$ represents the number of GNN layers, where $\mathbf{h}_i^{(0)} = \mathbf{x}_i$, and $\mathbf{h}_i^{(l)}(1 \leq l \leq L)$ denotes the node embedding of $v_i$ at the $l$-th layer. $\mathcal{N}(v_i)$ denotes the 1-hop neighbors of $v_i$, and $\texttt{AGGR}(\cdot)$ and $\texttt{COMB}(\cdot)$ are used for aggregating neighborhood information and combining ego/neighbor-representations, respectively. Finally, a sum/mean pooling operation is commonly used for $\texttt{READOUT}$ function to obtain the graph-level embedding. While promising, the increasing volume of graph samples [55, 19, 56] poses significant computational challenges for both training and pre-training of GNNs. Efficiently accelerating graph-level training remains an unresolved issue.

**Data Pruning**   Current data pruning methods can be categorized as static or dynamic [23]. Static data pruning involves heuristic-based metrics or limited training to assess sample importance and perform pruning before formal training, like EL2N [18] and Influence-score [57]. On the other hand, dynamic data pruning dynamically selects different training samples during training [58, 23, 59], often achieving better results than static pruning. In the graph domain, attempts related to data pruning include edge-level sampling techniques like GraphSAGE [60] and GraphSAINT [61]. However, to the best of our knowledge, there is currently no method specially designed for graph-level data pruning, let alone one that can simultaneously improve the balance and robustness of GNNs.

**Imbalance in GNNs**   Deep imbalanced learning has been one of the significant challenges in deep learning [62]. The current mainstream research can be broadly categorized into three approaches: (1) *re-sampling* [63, 64, 65], which balances the number of samples from different classes; (2) *re-balancing* [66, 67, 68], which adjusts the loss values for samples from different classes; and (3) *post-hoc processing* [69], which shifts the model logits based on label frequencies. In the domain of graph learning, most efforts to address the imbalance issue focus on node-level classification imbalances [70, 71, 72], yet solutions targeting graph-level imbalance are relatively limited. Despite a preliminary attempt [40], which requires complex up-sampling and regrouping operations, there is still a need for a straightforward yet effective solution to graph imbalance issue.

**Robustness in GNNs**   As for robustness learning, many studies showcase graph classification is vulnerable to adversarial attacks [73, 74]. Given a set of training or test graphs, an attacker could perturb the graph structure [75] and/or node features to deceive a graph classifier into making incorrect predictions for the perturbed testing graph. Traditional empirical and certified defenses [76, 77, 78, 79] often involve complex designs and additional components. In this paper, we propose subtle adjustments during training, leveraging prototypes to enhance the robustness of graph training.

## 3 Methodology

### 3.1 Problem Formulation

In the classic scenario of graph-level training (not limited to specific tasks like graph classification, regression, or pre-training), given a graph dataset $\mathcal{D} = \{z_i\}_{i=1}^{|\mathcal{D}|} = \{(\mathcal{G}_i, \mathbf{Y}_i)\}_{i=1}^{|\mathcal{D}|}$, a GNN encoder is employed to extract graph-level embeddings $\mathbf{H} = \{\mathbf{h}_i\}_{i=1}^{|\mathcal{D}|}$ for each graph sample, which are then utilized for downstream tasks. The goal of $\texttt{GDeR}$ is to find an oracle function that changes with time (epochs) and can determine the current most representative, balanced, and denoised core subset $\mathcal{X}_t$:

$$\mathcal{X}_t = \mathcal{F}_{t-1}\left(\mathcal{D}, \{\mathbf{h}_i^{(t-1)}\}_{i=1}^{|\mathcal{D}|}\right), \tag{2}$$

where $\mathcal{F}_{t-1}$ is the selection function at the $(t-1)$-th epoch. Given a preset sparsity ratio $s\%$, the subset's volume is fixed as $|\mathcal{X}_t| = (1 - s)\% \times |\mathcal{D}|$.

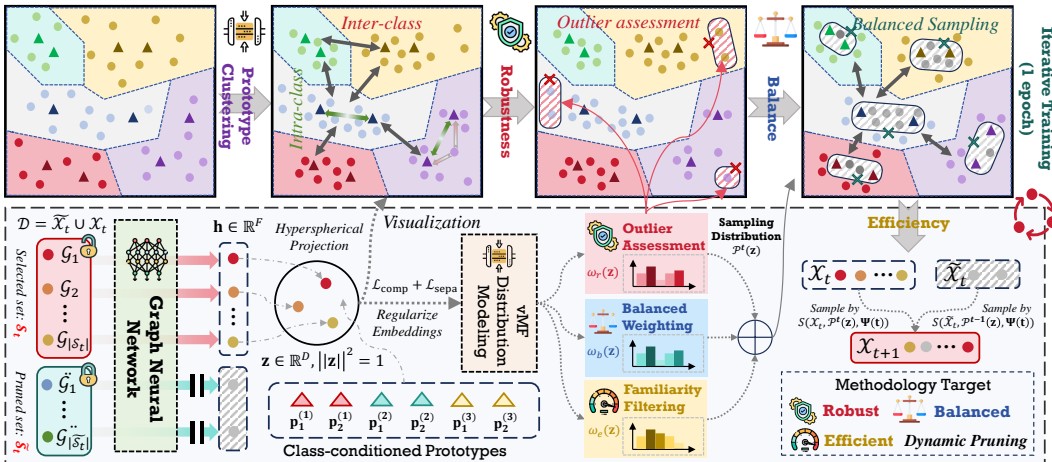

Figure 2: The overview of our proposed `GDeR`. `GDeR` comprises hypersphere projection, embedding space modeling, sampling distribution formatting, and the final dynamic sampling. We present the dynamic sample selection process of `GDeR` within one epoch.

## 3.2 Overview of the Proposed Method

As shown in Figure 2, given an arbitrary GNN, `GDeR` selects a training sample set $\mathcal{X}_t$ within a specified budget for each epoch. At the $t$-th epoch, after the GNN $f_\theta : \mathbf{X} \to \mathbb{R}^E$ outputs graph embeddings $\mathbf{h} \in \mathbb{R}^E$ from the input graph $\mathcal{G}_i$ with $\mathbf{h} = f_\theta(\mathcal{G}_i)$, these are projected into a hyperspherical embedding space via a *projector* $g_\phi : \mathbb{R}^E \to \mathbb{R}^D$. `GDeR` allocates a set of $M$ trainable prototypes $\mathbf{P}^c = \{\mathbf{p}_k^c\}_{k=1}^K$ for each class $c$, with associated losses used to shape the embedding space, ensuring inter-class separation and intra-class compactness. In this regularized space, `GDeR` formulates a sampling distribution by focusing on samples unfamiliar to the model, excluding those from the majority prototype cluster and with high outlier risk, thereby providing a subset of samples $S_{t+1}$ for the next epoch. Through this balanced and robust dynamic pruning mechanism, `GDeR` achieves unbiased graph representations at a significantly lower training cost than the full dataset.

## 3.3 Projection onto Hyperspherical Embedding Space

At the $t$-th epoch, `GDeR` maintains a subset $\mathcal{X}_t$ with a given budget, where $s\% = |\mathcal{X}_t|/|\mathcal{D}|$ is a constant, representing the dataset pruning ratio. Given the feature representations $\mathbf{H} \in \mathbb{R}^{|\mathcal{X}_t| \times E}$ output by $f_\theta$, we first project these features into a hyperspherical embedding space, denoted as $\mathbf{z}' = g_\phi(\mathbf{h}), \mathbf{z} = \mathbf{z}'/||\mathbf{z}'||_2$. This projection has been shown to be beneficial for compactly embedding samples of the same class [80, 81, 82]. The projected embeddings $\mathbf{z} \in \mathbb{R}^D$, which lie on the unit sphere ($||\mathbf{z}||^2 = 1$), can naturally be modeled using the von Mises-Fisher (vMF) distribution [80, 81]. Here, we first consider the graph classification scenario[1], in which we allocate $K$ prototypes $\mathbf{P}^c = \{\mathbf{p}_k^c\}_{k=1}^K$ for each class $c$ ($1 \leq c \leq C$). Following conventional practices in hyperspherical space modeling [83], we model a vMF distribution as the combination of a center prototype representation $\mathbf{p}_k$ and the concentration parameter $\kappa$:

$$p_D(\mathbf{z}; \mathbf{p}_k, \kappa) = Z_D(\kappa) \exp\left(\kappa \mathbf{p}_k^\top \mathbf{z}\right), \ Z_D(\kappa) = \frac{\kappa^{D/2-1}}{(2\pi)^{D/2} I_{D/2-1}(\kappa)}, \tag{3}$$

where $\kappa \geq 0$ denotes the tightness around the mean, $Z_D(\kappa)$ represents a normalization factor [83], $\exp\left(\kappa \mathbf{p}_k^\top \mathbf{z}\right)$ is called the angular distance and $I_v$ is the modified Bessel function of the first kind with order $v$. In our multi-prototype settings, we model the probability density of a graph embedding $\mathbf{z}_i$ in class $c$ as follows:

$$p(\mathbf{z}_i; \mathbf{P}^c, \kappa) = \sum_{k=1}^K Z_D(\kappa) \exp(\kappa \mathbf{p}_k^{c^\top} \mathbf{z}_i), \tag{4}$$

---

[1]The extension of `GDeR` to broader scenarios will be detailed in Section 3.5

Further, the embedding $\mathbf{z}_i$ is assigned to class $c$ with the normalized probability as shown above:

$$p(y_i = c \mid \mathbf{z}_i; \; \{\mathbf{P}^j, \kappa\}_{j=1}^C) = \frac{\sum_{k=1}^K Z_D(\kappa) \exp(\mathbf{p}_k^{c\top} \mathbf{z}_i / \tau)}{\sum_{j=1}^C \sum_{k'=1}^K Z_D(\kappa') \exp(\mathbf{p}_{k'}^{j\top} \mathbf{z}_i / \tau)}, \tag{5}$$

where $\tau$ is a temperature coefficient. Given that we have now allocated a corresponding class for each graph embedding, we aim to further encourage: ❶ *allocation correctness*, meaning that the allocation should be consistent with the ground truth label; ❷ *intra-class compactness*, meaning that graph embeddings should be close to the appropriate prototypes belonging to their own class; and ❸ *inter-class separateness*, meaning that graph embeddings should be distant from prototypes of other classes. To achieve ❶ and ❷ , we have designed the *compactness loss* below:

$$\mathcal{L}_{\text{comp}} = -\frac{1}{|\mathcal{X}_t|} \sum_{i=1}^{|\mathcal{X}_t|} \log \frac{\sum_{k=1}^K Z_D(\kappa) \exp(\mathbf{p}_k^{y_i\top} \mathbf{z}_i / \tau)}{\sum_{c=1}^C \sum_{k'=1}^K Z_D(\kappa') \exp(\mathbf{p}_{k'}^{y_i\top} \mathbf{z}_i / \tau)}, \tag{6}$$

where $y_i$ represents the class index for $\mathbf{z}_i$. Equation (6) is the maximum likelihood estimation of $\max_{\theta,\phi} \Pi_{i=1}^{|\mathcal{X}_t|} p(y_i = c | \mathbf{z}_i, \{\{\mathbf{p}_k^c, \kappa\}_{k=1}^K\}_{j=1}^C)$, which not only boosts the allocation correctness but also enforces graph embeddings to compactly surround the appropriate prototypes. Furthermore, to achieve ❸, namely encouraging inter-class separateness, we design the *separation loss*, optimizing large angular distances among different class prototypes:

$$\mathcal{L}_{\text{sepa}} = \frac{1}{C} \sum_{i=1}^C \log \frac{1}{C-1} \sum_{j=1}^C \mathbb{1}_{j \neq i} \sum_{k=1}^K \exp(\mathbf{p}_k^j \mathbf{z}_i / \tau) \tag{7}$$

where $\mathbb{1}(\cdot)$ is an indicator function. Through the above regularization, we obtain $C$ prototype clusters $\{\chi_c\}_{c=1}^C$, each composed of $K$ prototype centers $\{\mathbf{p}_k\}_{k=1}^K$ and surrounding sample sets $\{\mathbf{z}^{(C)}\}$. After modeling this hypersphere, we proceed with sample selection on the current subset $\mathcal{D}^{(t)}$.

### 3.4 Efficient, Balanced and Robust Graph Debugging

Traditional dynamic dataset pruning methods typically rely on loss-based metrics to select informative subsets [58, 23], which, however, can make the model more vulnerable to imbalance and malicious perturbation (as discussed in Section 1). In this subsection, while selecting a representative subset $\mathcal{D}^{(t)}$, we also intend to further ensure it is balanced and noise-free. Our first step is to locate samples that are at risk of being outliers in the embedding space. We propose using a *prototype-based Mahalanobis distance* to estimate the outlier risk of each graph sample:

$$\omega_{\text{r}}(\mathbf{z}_i) = -\min_c \left[ -\mathbb{1}_{y_i \neq c} \max_k \left[ (\mathbf{z}_i - \mathbf{p}_k^c)^\top \Sigma_k^{-1} (\mathbf{z}_i - \mathbf{p}_k^c) \right] \right], \tag{8}$$

where $\Sigma_k \in \mathbb{R}^{K \times K}$ is the sample covariance of all the prototypes in class $c$. Equation (8) calculates the maximum distance of $\mathbf{z}_i$ to all prototypes within its class, which serves as a robust outlier detection metric [84]. Furthermore, we intend to evaluate the effectiveness of each sample. Given that the distance of an embedding from its cluster center has been shown to be a good indicator of the model's familiarity with it [85], we compute the distance of each graph sample to its class-specific prototypes as a familiarity metric:

$$\omega_{\text{e}}(\mathbf{z}_i) = \frac{\sum_{k=1}^K \text{dist}(\mathbf{p}_k^{y_i}, \mathbf{z}_i)}{\sum_{c=1}^C \sum_{k'=1}^K \mathbb{1}_{c \neq y_i} \text{dist}(\mathbf{p}_{k'}^{y_i}, \mathbf{z}_i)}, \tag{9}$$

which suggests that if a graph sample is significantly closer to its own prototypes and farther from those of other classes, the model is more familiar with it. We implement the distance function using the angular distance in Equation (3). When considering the data balancing issue, we formulate the balancing score for each sample $\mathbf{z}_i$ as follows:

$$\omega_{\text{b}}(\mathbf{z}_i) = \left| \left\{ \mathbf{z}_i | \min_k \text{dist}(\phi_{\mathbf{z}_i}(\mathbf{p}_k), \mathbf{z}_i) \right\} \right| / |\{\phi_{\mathbf{z}_i}(\chi)\}|, \tag{10}$$

where $\phi_{\mathbf{z}_i}(\mathbf{p}_k)$ denotes the closest prototype to $\mathbf{z}_i$, and $\phi_{\mathbf{z}_i}(\chi)$ denotes the prototype cluster that $\mathbf{z}_i$ currently belongs to. Equation (10) evaluates whether the graph sample $\mathbf{z}_i$ belongs to a minority from a prototype-cluster perspective. Finally, we assign sampling probabilities to all samples in $\mathcal{D}^{(t)}$:

$$\omega(\mathbf{z}_i) = \frac{\omega_{\mathrm{e}}^{\sigma}(\mathbf{z}_i)}{(\omega_{\mathrm{r}}^{\sigma}(\mathbf{z}_i) + \epsilon) \cdot (\omega_{\mathrm{b}}^{\sigma}(\mathbf{z}_i) + \epsilon)}, \tag{11}$$

where $(\cdot)^{\sigma}$ represents the Sigmoid transformation. Equation (11) is designed to sample with higher probability those samples that the model is less familiar with, have a lower outlier risk, and belong to a minority group. Now, at the $t$-th epoch, we obtain the final sampling probability distribution $\mathcal{P}^{(t)}(\mathbf{z}) : \int_{\mathbf{z} \in \mathcal{X}_t} \frac{\omega(\mathbf{z})}{\int_{\mathbf{z}} \omega(\mathbf{z}) \, d\mathbf{z}} \, d\mathbf{z}$. Recall that we have $(1 - s)\%$ of samples pruned in the $t$-th epoch, *i.e.*, $\tilde{\mathcal{X}}_t = \mathcal{D} \setminus \mathcal{X}_t$. For $\tilde{\mathcal{X}}_t$, we use the probability distribution $\mathcal{P}^{(t-1)}(\mathbf{z})$ from the $(t-1)$-th epoch [2]. Specifically, we formulate GDeR's coreset sampling function $\mathcal{F}_t$ in Equation (2) as follows:

$$\mathcal{F}_t(\mathcal{D}, \mathbf{H}) = S\left(\mathcal{X}_t, \mathcal{P}^{(t)}(\mathbf{z}), \Psi(t)\right) \bigcup S\left(\tilde{\mathcal{X}}_t, \mathcal{P}^{(t-1)}(\mathbf{z}), \tilde{\Psi}(t)\right), \tag{12}$$

where $\mathcal{F}_t$ outputs the selected samples $\mathcal{X}_{t+1}$ for the next epoch's training, $S(\mathcal{X}, \mathcal{P}, N)$ is a sampling operator that samples $N$ samples from $\mathcal{X}$ with probability distribution $\mathcal{P}$, and $\Psi(t)$ ($\tilde{\Psi}(t)$) is the scheduler function (with implementation placed in Appendix B.5) that control the number of samples drawn from $\mathcal{X}_t$ ($\tilde{\mathcal{X}}_t$), respectively, subject to the given budget $\Psi(t) + \tilde{\Psi}(t) = |\mathcal{D}| \times s\% = |\mathcal{X}_t|$.

### 3.5 Optimization and Extension

**Optimization**    Aside from the original task-specific loss of GNN training denoted as $\mathcal{L}_{\mathrm{task}}$, GDeR has additionally introduced $\mathcal{L}_{\mathrm{comp}}$ and $\mathcal{L}_{\mathrm{sepa}}$. The overall training objective of GDeR is formulated as:

$$\mathcal{L}_{\mathrm{GDeR}} = \mathcal{L}_{\mathrm{task}} + \lambda_1 \cdot \mathcal{L}_{\mathrm{comp}} + \lambda_2 \cdot \mathcal{L}_{\mathrm{sepa}}, \tag{13}$$

where $\lambda_1$ and $\lambda_2$ are co-efficient adjusting the relative importance of two losses. We conclude the algorithm workflow table of GDeR in Appendix C.

**Extension**    Finally, we advocate that GDeR is not limited to graph classification but can also be seamlessly adapted to tasks such as graph regression and graph pre-training. The key distinction between these tasks and graph classification is that each graph sample does not have a ground truth class index, which makes ground truth class-based calculations, such as those in Equations (6) and (7), infeasible. One straightforward approach is to manually set $M$ virtual classes, using the class assigned by Equation (5) as the graph sample's current class. However, this may result in prototypes and hyperspherical embeddings that do not accurately reflect the underlying clustering distribution [86]. To address this, we leverage ProtNCELoss [43] as a self-supervised signal, providing a more reliable reflection of the data's structure. Detailed implementation can be found in Appendix D.

## 4 Experiments

In this section, we conduct extensive experiments to answer the following research questions: (**RQ1**) Can GDeR effectively boost GNN efficiency (under both supervised and unsupervised settings)? (**RQ2**) Does GDeR genuinely accelerate the GNN training? (**RQ3**) Can GDeR help alleviate graph imbalance? (**RQ4**) Can GDeR aid in robust GNN training?

### 4.1 Experiment Setup

**Datasets and Backbones**    We test GDeR on two widely-used datasets, MUTAG [38] and DHFR [87]; two OGB large-scale datasets, OGBG-MOLHIV and OGBG-MOLPBCA [88]; one large-scale chemical compound dataset ZINC [89]. Following [40], we adopt a 25%/25%/50% train/validation/test random split for the MUTAG and DHFR under imbalanced scenarios and 80%/10%/10% under normal and biased scenarios, both reporting results across 20 data splits. For OGBG-MOLHIV and OGBG-MOLPBCA, we use the official splits provided by [88]. For ZINC, we follow the splits specified in [90]. We choose three representative GNNs, including GCN [91], PNA [92] and GraphGPS [90]. Detailed dataset and backbone settings are in Appendices B.1 and B.2.

**Parameter Configurations**    The hyperparameters in GDeR include the temperature coefficient $\tau$, prototype count $K$, loss-specific coefficient $\lambda_1$ and $\lambda_2$. Practically, we uniformly set $K = 2$, and tune the other three by grid searching: $\tau \in \{1e-3, 1e-4, 1e-5\}$, $\lambda \in \{1e-1, 5e-1\}$, $\lambda \in \{1e-1, 1e-5\}$. Detailed ablation study on hyperparameters is placed in Section 4.5.

---

[2] For the first epoch, we set $\mathcal{P}^{(t-1)}(\mathbf{z})$ as uniform distribution.

Table 1: Performance comparison to state-of-the-art dataset pruning methods when remaining $\{20\%, 30\%, 50\%, 70\%\}$ of the full set. All methods are trained using **PNA**, and the reported metrics represent the average of **five runs**.

| Dataset | | OGBG-MOLHIV (ROC-AUC ↑) | | | | OGBG-MOLPCBA (AP ↑) | | | |
|---|---|---|---|---|---|---|---|---|---|
| Remaining Ratio % | | 20 | 30 | 50 | 70 | 20 | 30 | 50 | 70 |
| Static | Hard Random | $72.1_{\downarrow 4.2}$ | $72.4_{\downarrow 3.9}$ | $73.5_{\downarrow 2.8}$ | $75.6_{\downarrow 0.7}$ | $20.5_{\downarrow 7.6}$ | $22.9_{\downarrow 5.2}$ | $24.7_{\downarrow 3.4}$ | $28.0_{\downarrow 0.1}$ |
| | CD [93] | $71.9_{\downarrow 4.4}$ | $72.6_{\downarrow 3.7}$ | $73.8_{\downarrow 2.5}$ | $75.9_{\downarrow 0.4}$ | $19.8_{\downarrow 8.3}$ | $22.6_{\downarrow 5.5}$ | $23.7_{\downarrow 4.4}$ | $27.8_{\downarrow 0.3}$ |
| | Herding [94] | $63.0_{\downarrow 13.3}$ | $64.9_{\downarrow 11.4}$ | $66.8_{\downarrow 9.5}$ | $75.2_{\downarrow 1.1}$ | $12.4_{\downarrow 15.7}$ | $14.0_{\downarrow 14.1}$ | $15.5_{\downarrow 12.6}$ | $21.8_{\downarrow 6.3}$ |
| | K-Means [95] | $61.5_{\downarrow 14.8}$ | $65.9_{\downarrow 10.4}$ | $69.5_{\downarrow 6.8}$ | $74.7_{\downarrow 2.6}$ | $18.5_{\downarrow 9.6}$ | $23.4_{\downarrow 4.7}$ | $23.2_{\downarrow 4.9}$ | $27.6_{\downarrow 0.5}$ |
| | Least Confidence [96] | $72.1_{\downarrow 4.2}$ | $72.4_{\downarrow 3.9}$ | $75.6_{\downarrow 0.7}$ | $75.9_{\downarrow 0.4}$ | $21.0_{\downarrow 7.1}$ | $23.4_{\downarrow 4.7}$ | $25.0_{\downarrow 3.1}$ | $27.8_{\downarrow 0.3}$ |
| | Margin [96] | $72.9_{\downarrow 3.4}$ | $71.3_{\downarrow 5.0}$ | $75.1_{\downarrow 1.2}$ | $76.0_{\downarrow 0.3}$ | $20.2_{\downarrow 7.9}$ | $23.3_{\downarrow 4.8}$ | $25.0_{\downarrow 3.1}$ | $28.3_{\uparrow 0.2}$ |
| | Forgetting [33] | $72.6_{\downarrow 3.7}$ | $73.0_{\downarrow 3.3}$ | $73.9_{\downarrow 2.4}$ | $75.7_{\downarrow 0.6}$ | $20.7_{\downarrow 7.4}$ | $23.1_{\downarrow 5.0}$ | $24.1_{\downarrow 4.0}$ | $27.9_{\downarrow 0.2}$ |
| | GraNd-4 [18] | $68.5_{\downarrow 7.8}$ | $72.7_{\downarrow 3.6}$ | $73.8_{\downarrow 2.5}$ | $75.7_{\downarrow 0.6}$ | $20.2_{\downarrow 7.9}$ | $22.9_{\downarrow 5.2}$ | $25.0_{\downarrow 3.1}$ | $28.0_{\downarrow 0.1}$ |
| | GraNd-20 [18] | $74.7_{\downarrow 1.6}$ | $74.0_{\downarrow 2.3}$ | $74.9_{\downarrow 1.4}$ | $75.9_{\downarrow 0.4}$ | $21.2_{\downarrow 6.9}$ | $23.8_{\downarrow 4.3}$ | $24.9_{\downarrow 3.2}$ | $27.8_{\downarrow 0.3}$ |
| | DeepFool [97] | $71.9_{\downarrow 4.4}$ | $72.5_{\downarrow 3.8}$ | $73.0_{\downarrow 3.3}$ | $75.6_{\downarrow 0.7}$ | $19.3_{\downarrow 8.8}$ | $22.7_{\downarrow 5.4}$ | $24.0_{\downarrow 4.1}$ | $27.7_{\downarrow 0.4}$ |
| | Craig [98] | $71.8_{\downarrow 4.5}$ | $72.3_{\downarrow 4.0}$ | $73.5_{\downarrow 2.8}$ | $76.0_{\downarrow 0.3}$ | $20.5_{\downarrow 7.6}$ | $23.1_{\downarrow 5.0}$ | $24.7_{\downarrow 3.4}$ | $27.8_{\downarrow 0.3}$ |
| | Glister [99] | $73.3_{\downarrow 3.0}$ | $74.4_{\downarrow 2.9}$ | $75.0_{\downarrow 1.3}$ | $76.2_{\downarrow 0.1}$ | $20.6_{\downarrow 7.5}$ | $23.4_{\downarrow 4.7}$ | $25.0_{\downarrow 3.1}$ | $27.9_{\downarrow 0.2}$ |
| | Influence [57] | $71.5_{\downarrow 4.8}$ | $72.7_{\downarrow 3.6}$ | $73.5_{\downarrow 2.8}$ | $75.2_{\downarrow 1.1}$ | $19.7_{\downarrow 8.4}$ | $22.3_{\downarrow 5.8}$ | $23.9_{\downarrow 4.2}$ | $27.2_{\downarrow 0.9}$ |
| | EL2N-2 [33] | $73.0_{\downarrow 3.3}$ | $74.5_{\downarrow 1.8}$ | $75.0_{\downarrow 1.3}$ | $76.1_{\downarrow 0.2}$ | $20.9_{\downarrow 7.2}$ | $23.5_{\downarrow 4.6}$ | $24.3_{\downarrow 3.8}$ | $27.6_{\downarrow 0.5}$ |
| | DP [100] | $72.1_{\downarrow 4.2}$ | $73.5_{\downarrow 2.8}$ | $74.7_{\downarrow 1.6}$ | $76.0_{\downarrow 0.3}$ | $20.0_{\downarrow 8.1}$ | $22.7_{\downarrow 5.4}$ | $24.6_{\downarrow 3.5}$ | $27.7_{\downarrow 0.4}$ |
| Dynamic | Soft Random | $74.3_{\downarrow 2.0}$ | $73.9_{\downarrow 2.4}$ | $76.1_{\downarrow 0.2}$ | $76.2_{\downarrow 0.1}$ | $22.7_{\downarrow 5.4}$ | $24.8_{\downarrow 3.3}$ | $27.0_{\downarrow 1.1}$ | $27.8_{\downarrow 0.3}$ |
| | $\epsilon$-greedy [58] | $73.8_{\downarrow 2.5}$ | $73.6_{\downarrow 2.7}$ | $75.6_{\downarrow 0.7}$ | $76.2_{\downarrow 0.1}$ | $24.0_{\downarrow 4.1}$ | $25.3_{\downarrow 2.8}$ | $27.1_{\downarrow 1.0}$ | $27.6_{\downarrow 0.5}$ |
| | UCB [58] | $73.8_{\downarrow 2.5}$ | $73.7_{\downarrow 2.6}$ | $75.0_{\downarrow 1.3}$ | $75.8_{\downarrow 0.5}$ | $23.9_{\downarrow 4.2}$ | $25.8_{\downarrow 2.3}$ | $26.6_{\downarrow 1.5}$ | $28.1_{\uparrow 0.0}$ |
| | InfoBatch [23] | $74.1_{\downarrow 2.2}$ | $74.0_{\downarrow 2.3}$ | $76.3_{\uparrow 0.0}$ | $76.3_{\uparrow 0.0}$ | $24.1_{\downarrow 4.0}$ | $24.8_{\downarrow 3.3}$ | $27.3_{\downarrow 0.8}$ | $28.3_{\uparrow 0.2}$ |
| | GDeR | $\mathbf{75.8}_{\downarrow 0.5}$ | $\mathbf{76.0}_{\downarrow 0.3}$ | $\mathbf{76.4}_{\uparrow 0.1}$ | $\mathbf{76.8}_{\uparrow 0.5}$ | $\mathbf{24.8}_{\downarrow 3.3}$ | $\mathbf{26.0}_{\downarrow 2.1}$ | $\mathbf{28.0}_{\downarrow 0.1}$ | $\mathbf{28.5}_{\uparrow 0.4}$ |
| | Whole Dataset | | $76.3_{\pm 0.9}$ | | | | $28.1_{\pm 0.3}$ | | |

## 4.2 GDeR makes GNN training way faster

To answer **RQ1** and **RQ2**, we comprehensively compare GDeR with **fourteen** widely-used static pruning methods and **three** dynamic pruning methods, as outlined in Table 1, with more detailed explanations in Appendix B.3. Following [23], we add hard random and soft random pruning as baselines for a more comprehensive comparison. Specifically, we set the dataset remaining ratio $(1-s)\% \in \{20\%, 30\%, 50\%, 70\%\}$. The performance results are shown in Tables 1, 2 and 7 and the efficiency comparisons are in Figure 3. Our observations (**Obs.**) are as summarized follows:

**Obs.❶ GDeR achieves maximum graph pruning with performance guarantees.** As shown in Tables 1 and 2, GDeR consistently outperforms both static or dynamic baselines under various pruning ratios. For instance, on OGBG-MOLHIV+PNA, GDeR experiences only a $0.5\%$ performance decay even with $80\%$ pruning, surpassing the current state-of-the-art method InfoBatch, which suffers a $1.7\%$ decay. When pruning $50\%$ and $30\%$ of the data, GDeR even achieves performance improvements of $0.1\%$ and $0.5\%$, respectively.

**Obs.❷ The degree of redundancy varies across different datasets.** We observe that OGBG-MOLPCBA is more sensitive to pruning than OGBG-MOLHIV, which suggests the degree of redundancy varies between datasets. For example, when pruning $80\%$ of the data, GraphGPS on OGBG-MOLPCBA exhibits a performance decay ranging between $3.5\% \sim 13.9\%$, significantly higher than the $2.5\% \sim 11.5\%$ decay observed on OGBG-MOLHIV. However, as the remaining ratio increases, GDeR quickly recovers and surpasses the full dataset performance by $0.2\%$ at the $50\%$ pruning level.

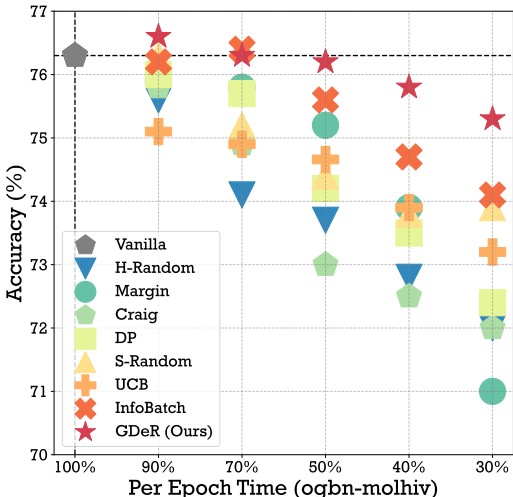

Figure 3: The trade-off between per epoch time and ROC-AUC (%) of data pruning methods. Specifically, we report the test performance when pruning methods achieve per epoch times of $\{90\%, 70\%, 50\%, 40\%, 30\%\}$ of the full dataset training time. "Vanilla" denotes the original GNN backbone without any data pruning.

Table 2: Performance comparison to state-of-the-art dataset pruning methods. All methods are trained using **GraphGPS**, and the reported metrics represent the average of **five runs**.

| Dataset | OGBG-MOLHIV (ROC-AUC ↑) | | | | OGBG-MOLPCBA (AP ↑) | | | |
|---|---|---|---|---|---|---|---|---|
| Remaining Ratio % | 20 | 30 | 50 | 70 | 20 | 30 | 50 | 70 |
| **Static** | | | | | | | | |
| Random | $69.3_{\downarrow 9.4}$ | $72.7_{\downarrow 6.0}$ | $73.4_{\downarrow 5.3}$ | $75.6_{\downarrow 3.1}$ | $19.4_{\downarrow 7.8}$ | $21.7_{\downarrow 5.5}$ | $23.9_{\downarrow 3.3}$ | $26.3_{\downarrow 0.9}$ |
| CD [93] | $72.6_{\downarrow 6.1}$ | $73.0_{\downarrow 5.7}$ | $75.3_{\downarrow 3.4}$ | $76.7_{\downarrow 2.0}$ | $18.0_{\downarrow 9.2}$ | $20.7_{\downarrow 6.5}$ | $21.7_{\downarrow 5.5}$ | $26.4_{\downarrow 0.8}$ |
| Herding [94] | $69.5_{\downarrow 9.2}$ | $73.3_{\downarrow 5.4}$ | $74.5_{\downarrow 4.2}$ | $75.9_{\downarrow 2.8}$ | $13.3_{\downarrow 13.9}$ | $14.0_{\downarrow 13.2}$ | $17.8_{\downarrow 9.4}$ | $23.0_{\downarrow 4.2}$ |
| K-Center [95] | $67.2_{\downarrow 11.5}$ | $70.8_{\downarrow 7.9}$ | $72.6_{\downarrow 6.1}$ | $73.9_{\downarrow 4.8}$ | $16.9_{\downarrow 10.3}$ | $19.4_{\downarrow 7.8}$ | $22.8_{\downarrow 4.4}$ | $26.1_{\downarrow 1.1}$ |
| Least Confidence [96] | $73.9_{\downarrow 4.8}$ | $74.2_{\downarrow 4.5}$ | $75.8_{\downarrow 2.9}$ | $77.3_{\downarrow 1.4}$ | $19.4_{\downarrow 7.6}$ | $21.9_{\downarrow 5.3}$ | $23.5_{\downarrow 3.7}$ | $26.0_{\downarrow 1.2}$ |
| Margin [96] | $74.0_{\downarrow 4.7}$ | $74.4_{\downarrow 4.3}$ | $75.8_{\downarrow 2.9}$ | $77.5_{\downarrow 1.2}$ | $18.8_{\downarrow 8.4}$ | $21.5_{\downarrow 5.7}$ | $23.9_{\downarrow 3.3}$ | $27.0_{\downarrow 0.2}$ |
| Forgetting [33] | $74.2_{\downarrow 4.5}$ | $74.8_{\downarrow 3.9}$ | $75.6_{\downarrow 3.1}$ | $76.9_{\downarrow 1.8}$ | $18.3_{\downarrow 9.9}$ | $21.9_{\downarrow 5.3}$ | $23.3_{\downarrow 3.9}$ | $26.8_{\downarrow 0.4}$ |
| GraNd-4 [18] | $73.8_{\downarrow 4.9}$ | $74.2_{\downarrow 4.5}$ | $75.3_{\downarrow 3.4}$ | $77.5_{\downarrow 1.2}$ | $18.0_{\downarrow 9.2}$ | $21.3_{\downarrow 5.9}$ | $23.6_{\downarrow 3.6}$ | $26.9_{\downarrow 0.3}$ |
| DeepFool [97] | $72.2_{\downarrow 6.5}$ | $73.3_{\downarrow 5.4}$ | $74.9_{\downarrow 3.8}$ | $75.5_{\downarrow 3.2}$ | $17.6_{\downarrow 9.6}$ | $21.9_{\downarrow 5.3}$ | $23.2_{\downarrow 4.0}$ | $26.5_{\downarrow 0.7}$ |
| Craig [98] | $73.5_{\downarrow 5.2}$ | $74.4_{\downarrow 4.3}$ | $76.0_{\downarrow 2.7}$ | $77.9_{\downarrow 0.8}$ | $18.7_{\downarrow 8.5}$ | $22.7_{\downarrow 4.5}$ | $24.5_{\downarrow 2.7}$ | $27.1_{\downarrow 0.1}$ |
| Glister [99] | $73.6_{\downarrow 5.1}$ | $74.0_{\downarrow 4.7}$ | $75.8_{\downarrow 2.9}$ | $78.0_{\downarrow 0.7}$ | $19.9_{\downarrow 7.3}$ | $22.5_{\downarrow 4.7}$ | $24.8_{\downarrow 2.4}$ | $27.0_{\uparrow 0.2}$ |
| Influence [57] | $72.9_{\downarrow 5.8}$ | $73.7_{\downarrow 5.0}$ | $74.8_{\downarrow 3.9}$ | $77.4_{\downarrow 1.3}$ | $17.7_{\downarrow 9.5}$ | $21.9_{\downarrow 5.3}$ | $23.5_{\downarrow 3.7}$ | $26.6_{\downarrow 0.6}$ |
| EL2N-20 [33] | $74.0_{\downarrow 4.7}$ | $75.5_{\downarrow 3.2}$ | $76.9_{\downarrow 1.8}$ | $77.7_{\downarrow 1.0}$ | $19.1_{\downarrow 8.1}$ | $22.9_{\downarrow 4.3}$ | $24.0_{\downarrow 3.2}$ | $26.0_{\downarrow 1.2}$ |
| DP [100] | $72.0_{\downarrow 6.7}$ | $74.1_{\downarrow 4.6}$ | $76.0_{\downarrow 2.7}$ | $76.9_{\downarrow 1.8}$ | $19.6_{\downarrow 7.6}$ | $21.5_{\downarrow 5.7}$ | $24.9_{\downarrow 2.3}$ | $26.4_{\downarrow 0.8}$ |
| **Dynamic** | | | | | | | | |
| Soft Random | $74.0_{\downarrow 4.7}$ | $74.1_{\downarrow 4.6}$ | $74.4_{\downarrow 4.3}$ | $78.1_{\downarrow 0.6}$ | $21.5_{\downarrow 5.7}$ | $22.4_{\downarrow 4.8}$ | $26.0_{\downarrow 1.2}$ | $27.1_{\downarrow 0.1}$ |
| $\epsilon$-greedy [58] | $74.7_{\downarrow 4.0}$ | $74.9_{\downarrow 3.8}$ | $76.6_{\downarrow 2.1}$ | $78.6_{\downarrow 0.1}$ | $22.8_{\downarrow 4.4}$ | $23.1_{\downarrow 4.1}$ | $26.3_{\downarrow 0.9}$ | $27.0_{\downarrow 0.2}$ |
| UCB [58] | $75.5_{\downarrow 3.2}$ | $74.9_{\downarrow 3.8}$ | $76.3_{\downarrow 2.4}$ | $78.0_{\downarrow 0.7}$ | $23.7_{\downarrow 3.5}$ | $24.1_{\downarrow 3.1}$ | $26.5_{\downarrow 0.7}$ | $27.2_{\uparrow 0.0}$ |
| InfoBatch [23] | $75.0_{\downarrow 3.4}$ | $75.6_{\downarrow 3.1}$ | $77.8_{\downarrow 0.9}$ | $78.5_{\downarrow 0.2}$ | $23.5_{\downarrow 3.7}$ | $\mathbf{24.6}_{\downarrow 2.6}$ | $26.7_{\downarrow 0.5}$ | $27.2_{\uparrow 0.0}$ |
| GDeR | $\mathbf{76.5}_{\downarrow 2.2}$ | $\mathbf{76.9}_{\downarrow 1.8}$ | $\mathbf{78.7}_{\uparrow 0.0}$ | $\mathbf{79.1}_{\uparrow 0.4}$ | $\mathbf{23.4}_{\downarrow 3.8}$ | $24.5_{\downarrow 2.7}$ | $\mathbf{27.4}_{\uparrow 0.2}$ | $\mathbf{27.6}_{\uparrow 0.4}$ |
| Whole Dataset | $78.7_{\pm 1.1}$ | | | | $27.2_{\pm 0.2}$ | | | |

**Obs. ❸ GDeR can significantly accelerate GNN training.** Figure 3 illustrates the per-epoch time and corresponding performance of each pruning method compared to full dataset training on OGBG-MOLHIV+GraphGPS. It is evident that GDeR can achieve a $2.0\times$ speedup without any performance loss (corresponding to $50\%$ per-epoch time). Even with a significant $3.3\times$ speedup, GDeR only experiences a moderate drop of $0.9\%$, which is superior to baselines including InfoBatch by a margin of $1.1\% \sim 4.2\%$. Additionally, we observe from Table 7 that pretraining on ZINC with only $30\%$ of the data leads to a $1.53\%$ ROC-AUC improvement, with $2.81\times$ training time acceleration.

### 4.3 GDeR Mitigates Graph Imbalance

To answer **RQ3**, we tested GDeR in extremely imbalanced scenarios and compared its performance with other dynamic pruning methods. Following [40], we randomly set $25\%/25\%$ graphs as training/-validation sets and within each of them, we designate one class as the minority class and reduce the number of graphs for this class in the training set (while increasing the others) until the imbalance ratio reached 1:9, which creates an extremely imbalanced scenario. The reported metrics are the average of 50 different data splits to avoid bias from data splitting. We observe from Figure 4 that:

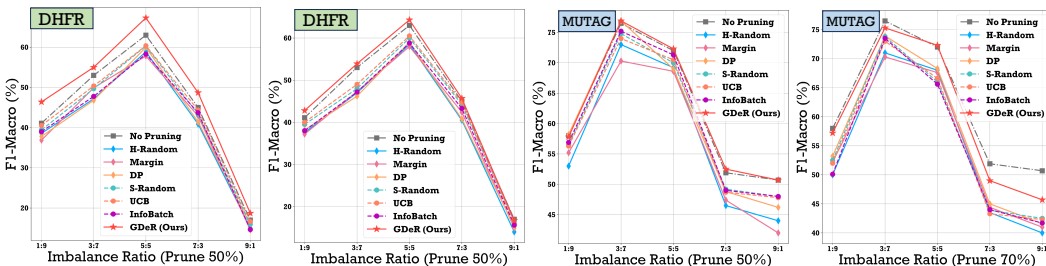

Figure 4: Performance comparison of different pruning methods across various imbalance ratios. We utilize MUTAG and DHFR datasets with GCN, and reported the metrics when adjusting the imbalance ratios among $\{1:9, 3:7, 5:5, 7:3, 9:1\}$. "No Pruning" denotes training GCN without dataset pruning.

**Obs. ❹ GDeR can effectively mitigate imbalance issues.** As observed in Figure 4, baseline pruning methods struggle to outperform "no-pruning" GCN, resulting in substantial losses in speedup efficacy. In contrast, GDeR offers a more meaningful pruning approach. For instance, on DHFR, pruning $50\%$ of the data results in a $4.3\%$ improvement in F1-Macro. This demonstrates that GDeR not only saves computational resources but also effectively mitigates data imbalance issues.

## 4.4 GDeR Aids in GNN Robustness

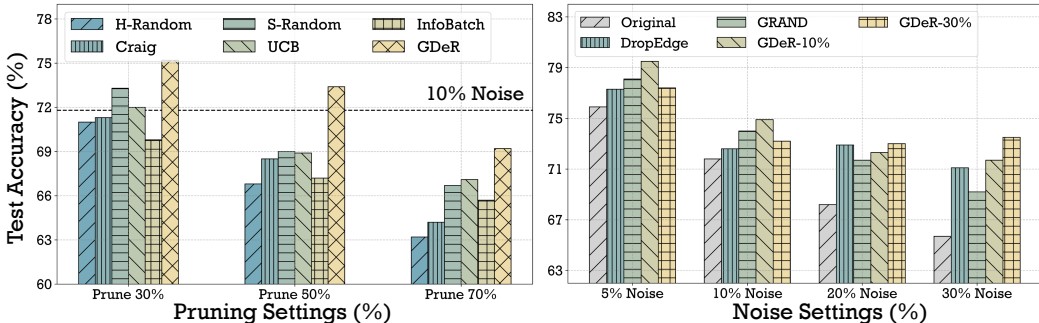

Figure 5: (*Left*) We report the performance of several top-performing pruning methods when perturbation noise is added to $10\%$ of the training set of MUTAG. The black dashed line represents the original GNN performance without pruning. (*Right*) We compare GDeR with DropEdge and GRAND under different noise settings, utilizing GDeR with pruning ratios of $10\%$ and $30\%$.

We divide **RQ4** into two sub-questions: (1) Is GDeR more robust to outlier perturbation compared to previous data pruning methods? (2) Can GDeR compete with mainstream methods designed to enhance GNN robustness? In practice, following [36], we introduce perturbations to $k\%$ of the graph samples in the training set by adding Gaussian noise to the node features of the selected graphs. We compare GDeR against both data pruning baselines and GNN robustness enhancement baselines. The experimental results are presented in Figure 5, and we observe:

**Obs. ❺ GDeR is a resource-saving GNN robustness booster.** From Figure 5 (*Left*), we observe that GDeR effectively counters noise perturbation, outperforming the GNN under outlier attacks at both $30\%$ and $50\%$ pruning rates. Notably, InfoBatch, which performed competitively in **RQ1**, suffers a significant performance drop ($2.0\% \sim 6.1\% \downarrow$) in this biased training scenario, which is likely due to its loss magnitude-based sample selection mechanism, inadvertently amplifying the negative impact of high-loss outlier samples on the model. From Figure 5 (*Right*), we conclude that GDeR performs as well as or better than current robust GNN plugins, and it shows the most significant improvement in accuracy, with increases of $3.6\%$ and $7.8\%$ at noise ratios of $5\%$ and $30\%$, respectively.

## 4.5 Ablation & Sensitivity Study

**Ablation Study** To evaluate the effectiveness of the different modules in GDeR, we propose three variants: (1) GDeR w/o $\omega_e$, (2) GDeR w/o $\omega_r$, and (3) GDeR w/o $\omega_b$. GDeR w/o $\omega_e$ represents removing $\omega_e$ from Equation (11), with the other two variants defined similarly. We observe from Table 3 that ❶ removing any component leads to a performance drop for GDeR, while removing $\omega_b$ in the imbalance scenario or $\omega_r$ in the biased scenario results in the most significant impact; ❷ GDeR w/o $\omega_e$ consistently underperforms across all scenarios, indicating that selecting highly representative samples is fundamental to the success of dynamic pruning methods.

| Setting | Normal | Imbalance | Baised |
|---|---|---|---|
| GDeR | $84.21_{\pm 3.40}$ | $76.32_{\pm 4.70}$ | $77.84_{\pm 2.70}$ |
| GDeR w/o $\omega_e$ | $79.77_{\pm 2.97}$ | $73.78_{\pm 2.96}$ | $75.60_{\pm 3.55}$ |
| GDeR w/o $\omega_r$ | $84.01_{\pm 3.09}$ | $76.21_{\pm 3.42}$ | $77.96_{\pm 3.18}$ |
| GDeR w/o $\omega_b$ | $83.46_{\pm 2.50}$ | $73.12_{\pm 2.50}$ | $75.02_{\pm 2.98}$ |

Table 3: Ablation study on GDeR and its three variants. "Imbalance" refers to setting the imbalance ratio to be $\{1 : 9\}$, and "Noisy" refers to adding $5\%$ noise to the training set. All metrics are reported under $30\%$ pruning ratio.

| Ratio ($s\%$) | Metric | $K = 1$ | $K = 2$ | $K = 4$ |
|---|---|---|---|---|
| 20% | Perf. | $75.8_{\pm 1.5}$ | $\mathbf{76.5}_{\pm 1.4}$ | $76.1_{\pm 0.9}$ |
| | Time | 15.32 | 16.44 | 17.16 |
| 50% | Perf. | $78.2_{\pm 1.4}$ | $78.7_{\pm 1.3}$ | $\mathbf{78.9}_{\pm 0.23}$ |
| | Time | 19.97 | 20.18 | 22.08 |
| 70% | Perf. | $81.19_{\pm 2.0}$ | $79.1_{\pm 1.9}$ | $\mathbf{79.2}_{\pm 2.2}$ |
| | Time | 26.19 | 31.30 | 39.55 |

Table 4: Sensitivity analysis on $K$. We report the ROCAUC (%) and per-epoch time (s) on OGBG-MOLHIV+GraphGPS.

**Sensitivity and Efficiency Analysis** We investigate the impact of $K$, on the performance and efficiency of GDeR. Specifically, we vary $K \in \{1, 2, 4\}$ on OGBG-MOLHIV+GraphGPS and observe changes in performance and per-epoch time. We observe from Table 4 that $K = 1$ leads to an under-learning of the hypersphere, resulting in consistently lower performance. While $K = 4$ shows a marginal performance gain compared to $K = 2$, for efficiency considerations, we opt for $K = 2$ across all experiments. Additionally, we observe that data pruning significantly saves per-epoch time, with $s = 20$ resulting in per-epoch times being $40\% \sim 60\%$ of those achieved with $s = 70$.

# 5 Conclusion & Future Work

In this work, we propose the graph training debugging concept and explore soft dataset pruning in the graph learning area for the first time. Particularly, we present a prototype-guided soft pruning method, termed GDeR, which initially establishes a well-modeled graph embedding hypersphere and subsequently samples *representative, balanced, and noise-free subsets* from this embedding space, debugging and troubleshooting graph processing. In the future, we plan to extend this concept to the CV realm, aiming to expedite the process of image training and provide efficient insights for the development of high-quality visual large-scale models.

## Acknowledgement

Dawei Cheng is supported by the National Natural Science Foundation of China (Grant No. 62102287). Yuxuan Liang is supported by the National Natural Science Foundation of China (No. 62402414), Guangzhou Municipal Science and Technology Project (No. 2023A03J0011), and Guangdong Provincial Key Lab of Integrated Communication, Sensing and Computation for Ubiquitous Internet of Things (No. 2023B1212010007).

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

# Appendices

## A   Notations

We conclude the commonly used notations throughout the manuscript in Table 5.

Table 5: The notations that are commonly used in the manuscript.

| Notation | Definition |
|---|---|
| $\mathcal{G} = \{\mathcal{V}, \mathcal{E}\} = \{\mathbf{A}, \mathbf{X}\}$ | Input graph |
| $\mathbf{A}$ | Input adjacency matrix |
| $\mathbf{X}$ | Node features |
| $\mathcal{D} = \{z_i\}_{i=1}^{|\mathcal{D}|} = \{(\mathcal{G}_i, \mathbf{Y}_i)\}_{i=1}^{|\mathcal{D}|}$ | Graph datasets |
| $\mathbf{h}_i$ | Graph embedding for $\mathcal{G}_i$ |
| $f_\theta$ | GNN encoder |
| $g_\phi$ | Feature projector |
| $\mathbf{P}^c = \{\mathbf{p}_k^c\}_{k=1}^K$ | Total $K$ Prototypes for class $c$ |
| $\mathcal{X}_t$ | Remained training set at the $t$-th epoch |
| $\tilde{\mathcal{X}}_t$ | Pruned training set at the $t$-th epoch |
| $\mathbf{z}_i \in \mathbb{R}^D$ | Projected embedding for $\mathcal{G}_i$ |
| $\omega_r(\mathbf{z}_i)$ | Outlier risk assessment metric |
| $\omega_e(\mathbf{z}_i)$ | Sample familiarity matric |
| $\omega_b(\mathbf{z}_i)$ | Sample balancing score |
| $\omega(\mathbf{z}_i)$ | Sampling probability for $\mathbf{z}_i$ |
| $\Psi(t)$ | Scheduler function that controls how many samples to choose from $\mathcal{X}_t$ |
| $\tilde{\Psi}(t)$ | Scheduler function that controls how many samples to choose from $\mathcal{X}_{t+1}$ |

## B   Experimental Details

### B.1   Dataset Details

The graph dataset details are summarized in Table 6.

Table 6: Graph datasets statistics.

| Dataset | #Graph | #Node | #Edge | #Classes | Metric |
|---|---|---|---|---|---|
| MUTAG | 188 | 17.93 | 19.79 | 2 | Accuracy/F1-macro |
| DHFR | 756 | 42.43 | 44.54 | 2 | Accuracy/F1-macro |
| OGBG-MOLHIV | 41,127 | 25.5 | 27.5 | 40 | ROC-AUC |
| OGBG-MOLPCBA | 437,929 | 26.0 | 28.1 | 2 | Average Precision |
| ZINC15 | 70,100 | 50.5 | 564.5 | 10 | Accuracy |

## B.2 Backbone Settings

We choose three representative GNNs, including one classic message-passing network GCN [91], one classical graph classification backbone PNA [92] and a graph transformer backbone GraphGPS [90]. For GCN, we simply set $layer\_num = 3$ and `hidden_dim` $= 128$. For PNA, we set $layer\_num = 4$, and `hidden_dim` $= 64$, `edge_dim` $= 16$. The rest configurations are the same as provided by [92] (https://github.com/lukecavabarrett/pna/blob/master/models/pytorch_geometric/example.py). For GraphGPS, we uniformly set `hidden_dim` $= 64$, `pe_dim` $= 8$ and utilize random walk encoding, performer [101], and GINE [102] as the positional encoding, attention module and the local convolutional module, respectively. The rest configurations are the same as provided by the PyTorch library (https://github.com/pyg-team/pytorch_geometric/blob/master/examples/graph_gps.py). All the experiments are conducted on NVIDIA Tesla V100 (32GB GPU), using PyTorch and PyTorch Geometric framework.

## B.3 Pruning Baselines

As for static pruning methods, we first introduce Hard Random, which conducts a random sample selection before training. Influence [57] and EL2N [18] are two classical static pruning methods that prune samples based on Influence-score and EL2N-score, respectively. DP [100] conducts pruning with consideration of generalization. Following the methodology of [23], we introduce a total of 13 static data pruning methods. These methods select a core set of data via predefined score functions or heuristic knowledge. Additionally, we introduce three dynamic pruning methods, including $\epsilon$-greedy [58], UCB [58], and InfoBatch [23]. Following [58, 23], we also introduce the dynamic pruning baseline, termed Soft Random, which conducts random selection in each epoch.

## B.4 Metrics

For MUTAG and DHFR, the metrics used vary across different scenarios. In the normal (Section 4.2) and biased (Section 4.4) scenarios, we use accuracy. However, in the imbalanced scenario (Section 4.3), accuracy does not faithfully reflect the performance for the minority group. Following previous works in imbalanced classification [71], we choose to use F1-macro, which computes the accuracy independently for each class and then takes the average, treating different classes equally. For OGBG-MOLHIV and OGBG-MOLPCBA, we use ROC-AUC and Average Precision (AP), following [88].

## B.5 Scheduler Function

$\Psi(t)$ and $\tilde{\Psi}(t)$ are scheduler functions that determine the proportions of samples in $\mathcal{X}_{t+1}$ originating from $\mathcal{X}_t$ and $\tilde{\mathcal{X}}_t$, respectively. For simplicity, we adopt the Inverse Power function [103]:

$$\Psi(t) = |\mathcal{X}_t| \cdot \varsigma \left(1 - \frac{t}{T}\right)^\kappa, \ \tilde{\Psi}(t) = |\mathcal{X}_t| - \Psi(t) \tag{14}$$

where $\varsigma$ denotes the initial ratio and $\kappa$ is the decay factor controlling the rate at which the ratio decreases over intervals. In practice, we uniformly set $\varsigma = 0.7$ and $\kappa = 2$.

# C Algorithm Workflow

The algorithm framework is presented in Algo. 1.

# D Extension of `GDeR`

In this section, we will explain how to extend `GDeR` beyond traditional graph classification tasks to more complex scenarios like graph regression and graph pre-training. As noted in Section 3.5, tasks such as graph pre-training do not have ground truth class indices, making direct application of `GDeR`, which relies on true class labels in Equations (6) and (7), infeasible.

A straightforward approach is to assign $C$ virtual classes, each with $K$ prototypes $\mathbf{P}^c = \{\mathbf{p}_k^c\}_{k=1}^K$. During prototype allocation, we use the probability distribution provided by Equation (5) to determine

---

**Algorithm 1:** Algorithm workflow of `GDeR`

---

**Input** : Graph datasets $\mathcal{D} = \{z_i\}_{i=1}^{|\mathcal{D}|} = \{(\mathcal{G}_i, \mathbf{Y}_i)\}_{i=1}^{|\mathcal{D}|}$, the number of epochs $T$, GNN encoder $f_\theta$, feature projector $g_\phi$,

Initialized $M$ prototypes $\{\mathbf{p}_k^{(C)}\}_{i=1}^M$ for class C

**for** epoch $t \leftarrow 1$ **to** $T$ **do**

    /* Extract graph-level embedding and Projection                        */

    $\mathcal{X}_t \leftarrow$ current training set.

    **for** sample index $i \leftarrow 1$ **to** $|\mathcal{X}_t|$ **do**

        Compute graph embedding $\mathbf{h}_i \leftarrow g_\theta(\mathcal{G}_i)$.

        Project graph embedding onto hypersphere by $\mathbf{z}_i = \mathbf{z}'/||\mathbf{z}'||_2, \mathbf{z}' = g_\theta(\mathbf{h}_i)$.

    **end**

    Determine which prototype cluster $\chi_c$ each graph sample $\mathbf{z}_i$ corresponds to;      ▷ Eq. 5

    /* Formatting sampling distribution                                    */

    Calculate the outlier score $\omega_r(\mathbf{z}_i)$ by prototype-based Mahalanobis distance;    ▷ Eq. 8

    Calculate the familiarity score $\omega_e(\mathbf{z}_i)$ based on prorotype-sample distance;    ▷ Eq. 9

    Calculate the balancing distribution $\omega_b$ based on cluster volume;    ▷ Eq. 10

    Formulate sampling distribution $\omega(\mathbf{z}_i) = \frac{\omega_e^\sigma(\mathbf{z}_i)}{(\omega_r^\sigma(\mathbf{z}_i)+\epsilon)\cdot(\omega_b^\sigma(\mathbf{z}_i)+\epsilon)}$;    ▷ Eq. 11

    /* Dataset sampling                                                 */

    Initialize the sample set for the $(t+1)$-th epoch $\mathcal{X}_{t+1} \leftarrow \emptyset$

    /* Sample from currently remained set                              */

    $\mathcal{X}_{t+1} \leftarrow \mathcal{X}_{t+1} + S\left(\mathcal{X}_t, \mathcal{P}^{(t-1)}(\mathbf{z}), \Psi(t)\right)$

    /* Sample from currently pruned set                               */

    $\mathcal{X}_{t+1} \leftarrow \mathcal{X}_{t+1} + S\left(\tilde{\mathcal{X}}_t, \mathcal{P}^{(t-1)}(\mathbf{z}), \tilde{\Psi}(t)\right)$

    /* Standard GNN training                                          */

    Compute loss $\mathcal{L}_{\text{GDeR}} = \mathcal{L}_{\text{task}} + \lambda_1 \cdot \mathcal{L}_{\text{comp}} + \lambda_2 \cdot \mathcal{L}_{\text{sepa}}$;    ▷ Eq. 13

    Backpropagate to update the GNN model $f_\theta$, projector $g_\phi$, and prototypes.

**end**

---

the class of each graph sample:

$$\tilde{y}_i = \arg\max_c p(y_i = c|\mathbf{z}_i, \{\mathbf{P}^j, \kappa\}_{j=1}^C). \tag{15}$$

We then substitute $\tilde{y}_i$ for $y_i$ in Equations (6) and (7), essentially emphasizing that the sample $\mathbf{z}_i$ should cluster tightly around its assigned prototype cluster $\chi_{\tilde{y}_i}$ and remain distant from other clusters. However, this approach is prone to error accumulation: if a sample is initially misclassified, $\mathcal{L}_{\text{comp}}$ and $\mathcal{L}_{\text{sepa}}$ will erroneously encourage it to continue moving in the wrong direction. To address this issue, we draw inspiration from previous practices in prototypical contrastive learning and leverage the prototypical contrastive loss [43]:

$$\mathcal{L}_{\text{contra}} = \sum_{i=1}^{|\mathcal{X}_t|} -\left(\frac{1}{C}\sum_{c=1}^C \log \frac{\sum_{k=1}^K \exp(\mathbf{z}_i \cdot \mathbf{p}_s^c / \phi_s^c)}{\sum_{j=0}^r \sum_{k=1}^K \exp(\mathbf{z}_i \cdot \mathbf{p}_j^c / \phi_j^m)}\right), \tag{16}$$

where $\phi$ calculates the concentration level of the feature distribution around a prototype as defined in [43]. Equation (16) has been shown to learn cluster distributions with high mutual information with ground truth labels in unsupervised settings. It encourages samples to migrate between clusters by measuring a concentration-weighted contrastive signal, rather than accumulating current errors. Thus, the overall objective of `GDeR` becomes:

$$\mathcal{L}'_{\text{GDeR}} = \mathcal{L}_{\text{task}} + \lambda_1 \cdot \mathcal{L}_{\text{comp}} + \lambda_2 \cdot \mathcal{L}_{\text{sepa}} + \lambda_3 \cdot \mathcal{L}_{\text{contra}}. \tag{17}$$

We applied this setting when extending `GDeR` to pre-training with GraphMAE on the ZINC dataset, with the experimental results in Table 7.

Table 7: Graph pre-training performance of `GDeR` on GraphMAE [47]+ZINC15 [104]. Following [47], the model is first pre-trained in 2 million unlabeled molecules sampled from the ZINC15, and then finetuned in 3 classification benchmark datasets contained in MoleculeNet [105].

| Remaining Ratio % | 30% | | | 50% | | | 70% | | |
|---|---|---|---|---|---|---|---|---|---|
| Dataset | BBBP | ToxCast | BACE | BBBP | ToxCast | BACE | BBBP | ToxCast | BACE |
| Original | 72.04 | 65.77 | 81.96 | 72.04 | 65.77 | 81.96 | 72.04 | 65.77 | 81.96 |
| +GDeR | 73.57 | 63.55 | 78.42 | 73.99 | 64.16 | 82.29 | 73.87 | 65.68 | 82.70 |
| Time consumption | 1.70 h | | | 2.58 h | | | 3.78 h | | |
| Training Speedup | $2.81\times$ | | | $1.86\times$ | | | $1.26\times$ | | |

The original pre-training time is 4.8 h.

Table 8: Performance comparison to state-of-the-art dataset pruning methods. All methods are trained using **PNA**, and the reported metrics represent the average of **twenty random runs** and different dataset splits.

| | Dataset | MUTAG (Accuracy ↑) | | | | DHFR (Accuracy ↑) | | | |
|---|---|---|---|---|---|---|---|---|---|
| | Remaining Ratio % | 20 | 30 | 50 | 70 | 20 | 30 | 50 | 70 |
| Static | Random | $85.3_{\downarrow 4.1}$ | $85.6_{\downarrow 3.8}$ | $86.7_{\downarrow 2.7}$ | $88.3_{\downarrow 1.1}$ | $72.3_{\downarrow 4.2}$ | $72.6_{\downarrow 3.9}$ | $73.7_{\downarrow 2.8}$ | $75.8_{\downarrow 0.7}$ |
| | CD [93] | $85.1_{\downarrow 4.3}$ | $85.8_{\downarrow 3.6}$ | $87.0_{\downarrow 2.4}$ | $88.1_{\downarrow 1.3}$ | $72.1_{\downarrow 4.4}$ | $72.8_{\downarrow 3.7}$ | $74.0_{\downarrow 2.5}$ | $76.1_{\downarrow 0.4}$ |
| | Herding [94] | $77.7_{\downarrow 11.7}$ | $79.6_{\downarrow 9.8}$ | $81.5_{\downarrow 7.9}$ | $87.9_{\downarrow 1.5}$ | $65.8_{\downarrow 10.7}$ | $67.6_{\downarrow 8.9}$ | $69.6_{\downarrow 6.9}$ | $73.6_{\downarrow 2.9}$ |
| | K-Center [95] | $76.2_{\downarrow 3.2}$ | $80.6_{\downarrow 8.8}$ | $84.2_{\downarrow 5.2}$ | $88.4_{\downarrow 1.0}$ | $64.2_{\downarrow 12.3}$ | $68.2_{\downarrow 8.3}$ | $70.6_{\downarrow 5.9}$ | $72.8_{\downarrow 3.7}$ |
| | Least Confidence [96] | $85.3_{\downarrow 4.1}$ | $85.6_{\downarrow 3.8}$ | $87.8_{\downarrow 1.6}$ | $88.3_{\downarrow 1.1}$ | $72.3_{\downarrow 4.2}$ | $72.6_{\downarrow 3.9}$ | $74.8_{\downarrow 1.7}$ | $76.1_{\downarrow 0.4}$ |
| | Margin [96] | $84.2_{\downarrow 5.2}$ | $84.5_{\downarrow 4.9}$ | $87.3_{\downarrow 2.1}$ | $88.4_{\downarrow 1.0}$ | $70.2_{\downarrow 6.3}$ | $71.5_{\downarrow 5.0}$ | $74.6_{\downarrow 1.9}$ | $75.6_{\downarrow 0.9}$ |
| | Forgetting [33] | $85.8_{\downarrow 3.6}$ | $86.2_{\downarrow 3.2}$ | $87.1_{\downarrow 2.3}$ | $88.4_{\downarrow 1.0}$ | $72.8_{\downarrow 3.7}$ | $73.2_{\downarrow 3.3}$ | $74.1_{\downarrow 2.4}$ | $76.0_{\downarrow 0.5}$ |
| | GraNd-4 [18] | $81.7_{\downarrow 7.7}$ | $85.9_{\downarrow 3.5}$ | $87.0_{\downarrow 2.4}$ | $88.2_{\downarrow 1.2}$ | $68.7_{\downarrow 7.8}$ | $72.9_{\downarrow 3.6}$ | $74.0_{\downarrow 2.5}$ | $75.6_{\downarrow 0.9}$ |
| | DeepFool [97] | $85.1_{\downarrow 4.3}$ | $85.6_{\downarrow 3.8}$ | $86.7_{\downarrow 2.7}$ | $88.1_{\downarrow 1.3}$ | $72.1_{\downarrow 4.4}$ | $72.7_{\downarrow 3.8}$ | $73.2_{\downarrow 3.3}$ | $75.8_{\downarrow 0.7}$ |
| | Craig [98] | $85.0_{\downarrow 4.4}$ | $85.4_{\downarrow 4.0}$ | $86.3_{\downarrow 3.1}$ | $88.2_{\downarrow 1.2}$ | $72.0_{\downarrow 4.5}$ | $72.5_{\downarrow 4.0}$ | $73.7_{\downarrow 2.8}$ | $76.2_{\downarrow 0.3}$ |
| | Glister [99] | $86.3_{\downarrow 3.1}$ | $86.8_{\downarrow 2.6}$ | $87.2_{\downarrow 2.2}$ | $88.4_{\downarrow 1.0}$ | $72.9_{\downarrow 3.6}$ | $73.3_{\downarrow 3.2}$ | $75.2_{\downarrow 1.3}$ | $76.4_{\downarrow 0.1}$ |
| | Influence [57] | $84.7_{\downarrow 4.7}$ | $85.9_{\downarrow 3.5}$ | $86.7_{\downarrow 2.7}$ | $88.1_{\downarrow 1.3}$ | $71.7_{\downarrow 4.8}$ | $72.9_{\downarrow 3.6}$ | $73.7_{\downarrow 2.8}$ | $75.4_{\downarrow 1.1}$ |
| | EL2N-2 [33] | $86.2_{\downarrow 3.2}$ | $87.1_{\downarrow 2.3}$ | $87.7_{\downarrow 1.7}$ | $88.2_{\downarrow 1.2}$ | $72.2_{\downarrow 4.3}$ | $73.9_{\downarrow 2.6}$ | $74.6_{\downarrow 1.9}$ | $75.0_{\downarrow 1.5}$ |
| | EL2N-20 [33] | $86.3_{\downarrow 3.1}$ | $87.1_{\downarrow 2.3}$ | $87.9_{\downarrow 1.5}$ | $88.3_{\downarrow 1.1}$ | $72.4_{\downarrow 4.1}$ | $73.0_{\downarrow 3.5}$ | $74.6_{\downarrow 1.9}$ | $76.2_{\downarrow 0.3}$ |
| | DP [100] | $85.3_{\downarrow 4.1}$ | $86.2_{\downarrow 3.2}$ | $87.4_{\downarrow 2.0}$ | $87.9_{\downarrow 1.5}$ | $71.3_{\downarrow 5.2}$ | $72.6_{\downarrow 3.9}$ | $74.0_{\downarrow 2.5}$ | $75.6_{\downarrow 0.9}$ |
| Dynamic | Random* | $87.0_{\downarrow 2.4}$ | $86.6_{\downarrow 2.8}$ | $88.8_{\downarrow 0.6}$ | $88.9_{\downarrow 0.5}$ | $73.4_{\downarrow 3.1}$ | $73.9_{\downarrow 2.6}$ | $74.8_{\downarrow 1.7}$ | $76.4_{\downarrow 0.1}$ |
| | $\epsilon$-greedy [58] | $86.5_{\downarrow 2.9}$ | $86.3_{\downarrow 3.1}$ | $88.3_{\downarrow 1.1}$ | $88.9_{\downarrow 0.5}$ | $73.1_{\downarrow 3.4}$ | $73.4_{\downarrow 3.1}$ | $74.1_{\downarrow 1.6}$ | $76.4_{\downarrow 0.1}$ |
| | UCB [58] | $86.5_{\downarrow 2.9}$ | $86.4_{\downarrow 3.0}$ | $87.7_{\downarrow 1.7}$ | $88.5_{\downarrow 0.9}$ | $73.1_{\downarrow 3.4}$ | $73.5_{\downarrow 3.0}$ | $74.0_{\downarrow 2.5}$ | $75.9_{\downarrow 0.6}$ |
| | InfoBatch [23] | $86.8_{\downarrow 2.6}$ | $86.7_{\downarrow 2.7}$ | $89.0_{\downarrow 0.4}$ | $89.3_{\downarrow 0.1}$ | $73.3_{\downarrow 3.2}$ | $73.7_{\downarrow 2.8}$ | $75.0_{\downarrow 1.5}$ | $76.3_{\downarrow 0.2}$ |
| | GDeR | $\mathbf{88.2}_{\downarrow 1.2}$ | $\mathbf{88.5}_{\downarrow 0.9}$ | $\mathbf{89.3}_{\downarrow 0.1}$ | $\mathbf{89.9}_{\uparrow 0.5}$ | $\mathbf{75.7}_{\downarrow 0.8}$ | $\mathbf{75.9}_{\downarrow 0.6}$ | $\mathbf{76.1}_{\downarrow 0.4}$ | $\mathbf{77.1}_{\uparrow 0.6}$ |
| | Whole Dataset | $89.4_{\pm 0.1}$ | | | | $76.5_{\pm 0.1}$ | | | |

# E  Additional Experimental Results

We place additional results about MUTAG and DHFR in Table 8, and the GraphMAE+ZINC pre-training results in Table 7.

# F  Supplementary Related Work

**Constrastive Learning and Prototypical learning**   Contrastive representation learning methods consider each sample as a unique class, aligning multiple views of the same input while distancing other samples. This significantly improves the discriminative power of the learned representations, allowing these methods to excel in learning robust feature representations across unsupervised [106, 107, 108, 109, 110], semi-supervised [111], and supervised settings [81]. The foundational properties and effectiveness of contrastive loss within hyperspherical space have been extensively studied [80, 112]. Other approaches focus on learning feature representations by modeling the relationships between samples and cluster centroids [113] or prototypes [114]. Building on contrastive learning, [115] incorporates prototypical learning, adding a contrastive mechanism between samples and prototypes obtained through offline clustering. PALM [82] utilizes prorotypical learning for

out-of-distribution (OOD) identification, which automatically identifies and dynamically updates prototypes, and assigns each sample to a subset of prototypes via reciprocal neighbor soft assignment weights. However, all these methods are not conducive to a more lightweight training burden, and our method is the first attempt at leveraging prototype learning for soft data pruning.

