# OpenReview forum: "GDeR: Safeguarding Efficiency, Balancing, and Robustness via Prototypical Graph Pruning"
_NeurIPS.cc/2024/Conference — NeurIPS 2024 poster_

### Official Review · Reviewer_zhEK · 2024-07-04

**Soundness:** 3
**Presentation:** 4
**Contribution:** 3
**Rating:** 8
**Confidence:** 4

**Summary:**

This paper introduces a novel training debugging concept aimed at enhancing efficiency, robustness, and balance during the graph training process. It employs trainable prototypes to dynamically select appropriate samples for each training iteration. The concept is innovative and intriguing. However, the experiments are confined to the graph domain, which raises questions about its generalizability. The authors might need to further elaborate on this issue in the text to address potential limitations.

**Strengths:**

1. Topic of large importance in the community given the direction of the field. The authors proposed training debugging, namely robust and unbiased soft pruning, is indeed novel and significant, especially in the context of research on large models where data collection is increasingly extensive. This approach addresses critical challenges in ensuring that these large-scale models train effectively and efficiently, mitigating issues that might otherwise go unnoticed due to the complexity and size of the data involved.

2. The article is well written and engaging, particularly excelling in the clarity of its experimental tables and diagrams. These elements contribute to a structure that quickly aids readers in understanding the contributions of the paper.

**Weaknesses:**

1. One of my main concerns is whether this concept is only applicable to graph training. If it is limited solely to graphs, the overall contribution of this work may not reach a particularly high standard. Especially relevant is whether its methodology can be transferred to datasets for CV or NLP.

2. Intuitively, the introduction of prototypes may potentially decrease the training speed of the original GNN backbone. Considering there are $K$ prototypes and $|D|$ training samples, each epoch would require the computation of similarity scores between samples and prototypes at $\mathcal{O}(K \times |D|)$ space complexity. This is bound to introduce additional overhead during backpropagation. The authors need to provide discussions both on complexity analysis and numerical comparisons to quantify the extra computational burden.

3. The authors should report the results at extremely high sparsity levels (e.g. dropping 90% of the samples). Can your method still behaves fairly well?

4. The authors should also compare their method with other techniques for data imbalance in Section 4.3. Data imbalance/long-tail distribution is a long-standing issue with many established solutions in CV. To name a few, focal Loss [1] and Dynamic Sampling [2].

[1] Focal loss for dense object detection
[2] Dynamic sampling in convolutional neural networks for imbalanced data classification

**Questions:**

1. Is the proposed method extendable to other data domains?
2. How does the trainable prototypes effect the training speed?
3. Can GDeR remain its performance even pruning a great propotion of training samples?
4. Can you compare GDeR with other data imbalance baselines?

**Limitations:**

Yes

---

> ### Author Rebuttal · Authors · 2024-08-06
>
> We sincerely thank you for your careful comments and thorough understanding of our paper! Here we give point-by-point responses to your comments and describe the revisions we made to address them.
>
> ---
> > **Weakness 1 & Question 1**: Is the proposed method extendable to other data domains?
>
> To address your concerns regarding the transferability of GDeR to other domains, we will first explain (1) how GDeR can be adapted with minor modifications to other data domains and (2) the performance of GDeR on CV datasets.
>
> **Minor Modification** On Lines 146-148, we mention that in each epoch, any GNN encoder encodes a graph sample $\mathcal{G}\_i$ into an embedding $\mathbf{h}\_i\in\mathbb{R}^E$, which is then projected into hyperspherical space via the projector $g_\phi: \mathbb{R}^E \rightarrow \mathbb{R}^D$. If we apply this to other data scenarios, such as ResNet+ImageNet training, ResNet would map an image $\mathbf{x}_i$ from ImageNet into a one-dimensional vector $\mathcal{F}(x_i)\in\mathbb{R}^E$ after pooling. We then simply project this $\mathcal{F}(x_i)$ into hyperspherical space, enabling the use of GDeR for data soft pruning in the same manner. Therefore, you can see that GDeR can easily extend to classical backbones in other domains (e.g., ViT, Swin Transformer) and datasets (e.g., CIFAR, ImageNet, COCO).
>
> **Performance Evaluation** We test GDeR's performance on two classic CV datasets, CIFAR-10 and ImageNet-1k, as shown in Tables A and B. As can be seen, even on the large-scale ImageNet, GDeR demonstrated consistent training speedup without performance loss.
>
> *Table A. Accuracy (%) comparison between Random, InfoBatch and GDeR on CIFAR-10+ResNet-18. Remaining ratio are set among {100%, 70%, 30%}.*
> |Remaining ratio|100|70|30|
> |-|-|-|-|
> |Random|95.69|94.88|90.2|
> |InfoBatch|95.69|95.36|94.92|
> |GDeR|95.69|**95.74**|**95.12**|
>
>
> *Table B. Accuracy (%) and Time (hour) comparison between UCB, InfoBatch and GDeR on ImageNet-1k+ResNet-50. Remaining ratio are set as 60%. Our results are tested on eight NIVIDIA Tesla A100 (80GB GPU).*
> |Remaining ratio|100||60||
> |-|-|-|-|-|
> |Metric|Acc|Wall-clock Time|Acc|Wall-clock Time|
> |InfoBatch|76.43|6.2h|76.49|3.74h|
> |GDeR|76.43|6.2h|**76.52**|3.89h|
>
>
> ---
> > **Weakness 2 & Question 2**: The authors need to provide discussions both on complexity analysis and numerical comparisons to quantify the extra computational burden.
>
> Following your suggestions, we provide (1) complexity analysis and (2) numerical comparisons to better evaluate the efficiency of GDeR.
>
> **Complexity Analysis** Taking vanilla GCN as an example, its memory complexity is $\mathcal{O}\left(|\mathcal{D}|\times(L\times |\mathcal{N}|\times E + L\times |\mathbf{\Theta}|\times E^2)\right)$, where $|\mathcal{D}|$ denotes the graph sample count, $L$ denotes the GNN layer count, $|\mathcal{N}|$ denotes the (average) number of nodes, and $E$ represents the hidden dimension. GDeR utilizes an MLP-based projector that maps the graph-level embedding into hyperspherical space, introducing an additional $\mathcal{O}(|\mathcal{D}|\times D)$, where $D$ represents the hyperspherical embedding dimension. Furthermore, each sample needs to compute distances to prototypes, which introduces $\mathcal{O}(|\mathcal{D}|\times |\mathbf{P}|)$, where $|\mathbf{P}|$ denotes the number of prototypes. Overall, the additional complexity introduced by GDeR is $\mathcal{O}(|\mathcal{D}|\times (|\mathbf{P}|+D))$, which is significantly smaller than the complexity of GCN itself. Next, we empirically evaluate the extra burden caused by GDeR.
>
> **Numerical Comparisons** We provide a comparison of wall-clock time for GDeR, random pruning, and InfoBatch under different pruning rates in the context of MUTAG+PNA, as well as wall-clock time on GraphGPS+Molhiv. It can be observed that although GDeR incurs additional computational overhead due to prototype-related calculations compared to random pruning, these costs are marginal and lead to significant performance improvements (2.8% on Molhiv).
>
> *Table C. Results on MUTAG+PNA.*
> |  Sparsity     | 0.2      | 0.3      | 0.5      | 0.7      |
> |-|-|-|- | -- |
> | Ours | 115.9506 | 118.4320 | 195.7631 | 241.3048 |
> | Random | 100.6895 | 100.0426 | 183.6579 | 227.6009 |
> | InfoBatch | 102.5114 | 114.5953  | 185.8063 | 231.3231 |
>
> *Table D. Results on  GraphGPS+Molhiv.*
> |  (prune 70%)| Wall-clock time| Performance|
> |-|-|-|
> |Random| 1607.29| 74.1|
> |InfoBatch| 1694.55| 75.6|
> |GDeR| 1844.56|**76.9**|
>
> ---
> > **Weakness 3 & Question 3**: The authors should report the results at extremely high sparsity levels (e.g. dropping 90% of the samples). Can your method still behaves fairly well?
>
> To address your concerns, we compare GDeR with several best-performing baselines under extremely sparse scenarios.
>
> *Table E. Results on MUTAG+PNA.*
> |Remaining Ratio (%)|10|15|20|
> |-|-|-|-|
> |Random|82.71±1.29|85.92±0.94|87.60±0.52|
> |UCB|83.11±0.87|85.70±0.81|86.52±0.60|
> |InfoBatch|83.42±0.93|86.33±0.42|86.8±0.83|
> |GDeR|**83.59**±1.03|**87.15**±0.59|**88.24**±0.72|
>
> ---
> > **Weakness 4 & Question 4**: The authors should also compare their method with other techniques for data imbalance in Section 4.3.
>
> Thank you for your insightful suggestions! Following your instructions, we have supplemented our experiments in imbalanced scenarios by comparing GDeR with two classic data-balancing plugins. As shown in Table 4, GDeR outperforms existing data balancing techniques across various imbalanced settings.
>
> *Table F. F1-Macro (%) comparison of GDeR, DynamicSample and Graph-level SMOTE on MUTAG+GCN. We fix the pruning ratio of GDeR to 20%.*
> |Imbalance Ratio|1:9|3:7|7:3|9:1|
> |-|-|-|-|-|
> |Original|58.07|76.52|51.99|50.75|
> |DynamicSample[1]|63.12|75.41|73.07|69.20|
> |Graph SMOTE[2]|65.26|77.87|77.68|73.48|
> |GDeR|71.81|79.10|77.96|75.70|
>
> [1] Dynamic sampling in convolutional neural networks for imbalanced data classification
>
> [2] Imbalanced Graph Classification via Graph-of-Graph Neural Networks

---

> > ### Comment · Reviewer_zhEK · 2024-08-09
> >
> > Thank you for your response. Introducing ImageNet experiments and complexity analysis is persuasive and effectively showcases GDeR's adaptability. Personally, I appreciate the contribution of this work, and with its proper packaging and open-sourcing, it can become a widely applicable training acceleration technique. Therefore, I have adjusted my rating accordingly.

---

> > > ### Author Response · Authors · 2024-08-09
> > >
> > > We extend our heartfelt thanks to Reviewer zhEK for their increased support and recognition of our paper! We are pleased that our revisions and rebuttal have addressed the concerns!

---

### Official Review · Reviewer_kNzQ · 2024-07-08

**Soundness:** 2
**Presentation:** 3
**Contribution:** 3
**Rating:** 6
**Confidence:** 3

**Summary:**

This paper presents a novel method for graph neural network training called Graph De-Redundancy. This method aims to enhance the efficiency, balance, and robustness of GNN training. It constructs a hyperspherical embedding space using trainable prototypes to maintain a balanced subset of the training data, addressing the computational and performance challenges posed by large-scale, imbalanced, and noisy datasets. The experiments demonstrate that GDeR can achieve or surpass the performance of full datasets with significantly fewer samples, achieving up to a 2.81× speedup and outperforming existing pruning methods in various scenarios.  Overall, this approach achieves an effective, balancing, and robust pruning method for graph datasets, showing the potential for efficient training of graph datasets and even general datasets.

**Strengths:**

1. The study gives a powerful solution that meets the multifaceted needs of balance, stability and efficiency, and has great potential for application especially on large-scale and unbalanced graph datasets.
2. Using a hyperspherical embedding space and trainable prototypes for dataset pruning in graph learning is quite novel. In addition, the authors give sound explanations and theoretical support for these methods.
3. The paper is fluently written, with clear explanations of the problem, methodology and results. Especially the graphs and visualisations in the paper are impressive.

**Weaknesses:**

1. How are the prototypes initialized? I didn't see a note about it from the Algo. 1. Does the initialization of prototypes have an impact on the regulariaztion of hyperspherical space?
2. Does changing the weights of different losses such as $\lambda_1$ in Eq.(13) have an impact on the model output?
3. As shown in Eq. (14), at the beginning of training,$\Psi(t)$ will be relatively large tend to retain the graph retained in the last iteration, as the training proceeds $\Psi(t)$ will tend to be close to 0, resulting in a greater tendency to completely replace the graph retained in the last iteration, making the graph retained in each iteration varies greatly in the last few epochs, can you explain this phenomenon?
4. There appears to be some typos in line 245. $\lambda$ misses a subscript.

**Questions:**

This work looks like it could seamlessly migrate to the CV domain as the authors state in Section 3.5, have you done any extension experiments accordingly?

**Limitations:**

The authors have adequately discussed the limitations.

---

> ### Author Rebuttal · Authors · 2024-08-06
>
> Thank you immensely for your time and efforts, as well as the helpful and constructive feedback! Here, we give point-by-point responses to your comments.
>
> ---
> > **Weakness 1**: How are the prototypes initialized? I didn't see a note about it from the Algo. 1. Does the initialization of prototypes have an impact on the regulariaztion of hyperspherical space?
>
> Thank you for your insightful inquiry! GDeR follows classic prototype learning practices like [1], randomly initializing prototype embeddings for each cluster. Although different random initializations may affect the subsequent regularization of the hypersphere, we respectfully note that this impact is minimal. As shown in Table 4, even when the number of prototypes per cluster $K=1$, the performance variance of GDeR is around $1-2\%$; as $K$ increases, the performance variance tends to decrease.
>
> [1] Attribute prototype network for zero-shot learning
>
> ---
> > **Weakness 2**: Does changing the weights of different losses such as $\lambda_1$ in Eq.(13) have an impact on the model output?
>
> We are happy to answer your question, and conduct a sensitivity on $\lambda_1$ and $\lambda_2$ on MUTAG+PNA, as shown in Table A. We can conclude that (1) too small $\lambda_1$ or $\lambda_2$ can lead to under-fitting of hypersphere and less reliable data pruning; (2) GDeR pruning performs best when $\lambda_1$ and $\lambda_2$ are kept within a similar magnitude. Overall, GDeR is not sensitive to these two parameters, and practitioners can achieve desirable training speedup without extensive parameter tuning.
>
> *Table A. Sensitivity analyis of $\lambda_1$ and $\lambda_2$ on MUTAG+PNA. The pruning ratio is set to 50%.*
> |$\lambda_1$\ $\lambda_2$|5e-2|1e-1|5e-1|1e-0|
> |-|-|-|-|-|
> |5e-2|88.72|88.54|88.66|88.38|
> |1e-1|89.11|89.37|89.21|88.56|
> |5e-1|88.80|88.24|89.17|89.15|
> |1e-0|88.03|88.79|89.07|89.12|
>
> ---
> > **Weakness 3**: As shown in Eq. (14), $\Psi(t)$ at the beginning of training,
>  will be relatively large tend to retain the graph retained in the last iteration, as the training proceeds $\Psi(t)$ will tend to be close to 0, resulting in a greater tendency to completely replace the graph retained in the last iteration, making the graph retained in each iteration varies greatly in the last few epochs, can you explain this phenomenon?
>
> We apologize for any confusion caused! In Eq. (14), $\Psi(t)$ and $\widetilde{\Psi(t)}$ should be swapped, such that as the training proceeds $\Psi(t)$ will tend to be close to 1, indicating that we should retain the sample set more in the last few epochs. This is because, at the early stages of GNN training, the randomly selected sub-dataset is likely to be highly noisy, necessitating more frequent swapping. By the end of the training, GDeR has progressively filtered out truly representative, unbiased, and balanced sub-datasets, thus reducing the need for sample swapping.
>
> ---
> > **Weakness 4**: There appears to be some typos in line 245. $\lambda$ misses a subscript.
>
> Thank you for your detailed comments! On Line 245, we conducted a grid search for $\lambda_1$ and $\lambda_2$, both ranging in {$1e-1, 5e-1$}.
>
> ---
> > **Question 1**: This work looks like it could seamlessly migrate to the CV domain as the authors state in Section 3.5, have you done any extension experiments accordingly?
>
> To address your concerns regarding the transferability of GDeR to other domains, we will first explain (1) how GDeR can be adapted with minor modifications to other data domains and (2) the performance of GDeR on CV datasets.
>
> **Minor Modification** On Lines 146-148, we mention that in each epoch, any GNN encoder encodes a graph sample $\mathcal{G}\_i$ into an embedding $\mathbf{h}\_i \in \mathbb{R}^{E}$, which is then projected into hyperspherical space via the projector $g_\phi: \mathbb{R}^E \rightarrow \mathbb{R}^D$. If we apply this to other data scenarios, such as ResNet+ImageNet training, ResNet would map an image $\mathbf{x}_i$ from ImageNet into a one-dimensional vector $\mathcal{F}(x_i)\in\mathbb{R}^E$ after pooling. We then simply project this $\mathcal{F}(x_i)$ into hyperspherical space, enabling the use of GDeR for data soft pruning in the same manner. Therefore, you can see that GDeR can easily extend to classical backbones in other domains (e.g., ViT, Swin Transformer) and datasets (e.g., CIFAR, ImageNet, COCO).
>
> **Performance Evaluation** We test GDeR's performance on two classic CV datasets, CIFAR-10 and ImageNet-1k, as shown in Tables B and C. As can be seen, even on the large-scale ImageNet, GDeR demonstrated consistent training speedup without performance loss.
>
> *Table 1. Accuracy (%) comparison between Random, InfoBatch and GDeR on CIFAR-10+ResNet-18. Remaining ratio are set among {100%, 70%, 30%}. Our results are tested on one NIVIDIA Tesla A100 (80GB GPU).*
> |Remaining ratio|100|70|30|
> |-|-|-|-|
> |Random|95.69|94.88|90.2|
> |InfoBatch|95.69|95.36|94.92|
> |GDeR|95.69|**95.74**|**95.12**|
>
>
> *Table 2. Accuracy (%) and Time (h) comparison between UCB, InfoBatch and GDeR on ImageNet-1k+ResNet-50. Remaining ratio are set as 60%. Our results are tested on eight NIVIDIA Tesla A100 (80GB GPU).*
> |Remaining ratio|100||60||
> |-|-|-|-|-|
> |Metric|Acc|Time|Acc|Time|
> |InfoBatch|76.43|6.2h|76.49|3.74h|
> |GDeR|76.43|6.2h|**76.52**|3.89h|

---

### Official Review · Reviewer_RKXd · 2024-07-08

**Soundness:** 3
**Presentation:** 3
**Contribution:** 3
**Rating:** 7
**Confidence:** 4

**Summary:**

This paper addresses the computational and memory challenges posed by large datasets in the training of graph neural networks (GNNs). The authors propose GDeR, a dynamic soft-pruning method that leverages trainable prototypes to regularize the graph embedding distribution. This approach aims to maintain a balanced and unbiased subset of the data during training, thereby enhancing the efficiency and robustness of the learning process.

**Strengths:**

(1) Soft dataset pruning for graph data remains underexplored for the past years, and GDER makes the first attempt.
(2) The paper reads very well, with straightforward motivation and well-organized methodology. I appreciate its visual aids (Fiure 2,3).
(3) The experimental results (particularly on GraphMAE+ZINC) are encouraging. It is demonstrated that GDER can save over 60% of the training wall-clock time without performance decay. I believe GDER has the potential to become a general acceleration operator/plugin for the graph learning community.

**Weaknesses:**

(1) The methodology is primarily organized around graph classification. Though the authors claim that GDER can be extended to graph regression, more experiments are needed to validate their claim in Section 4.2.
(2) In robust GNN training, the baselines compared are outdated. There are stronger robust GNN baselines [1,2,3].
(3) The authors merely test GDER with poisoning attack on node features. Is GDER capable of addresing evasion attack? Also, the experiments can be stronger if the authors use more advanced GNN attack methods like Mettack [4] or Nettack [5].


Minor:

I recommend the authors to move Table 7 to the main text. Pretraining GNNs present more computational burden, so its accelerating is more meaningful. Besides, On Lines 184 and 189, does $D^{t}$ refers to $X_t$?


[1] Adversarial robustness in graph neural networks: A Hamiltonian approach, NeurIPS 2023.
[2] A Simple and Yet Fairly Effective Defense for Graph Neural Networks, AAAI 2024
[3] Graphde: A generative framework for debiased learning and out-of-distribution detection on graphs, NeurIPS 2022
[4] Adversarial attacks on node embeddings via graph poisoning, ICML 2019
[5] Adversarial attacks on neural networks for graph data, KDD 2018

**Questions:**

(1) Figure 1 shows the sample label ditrbution using Infobatch, right? If so, what is the distribution like with GDER?
(2) Can you give a case study of the graph samples pruned by GDER? For examples, how many of them are with noise? how many of them are from the majority class?

**Limitations:**

The limiations are discussed in the paper.

---

> ### Author Rebuttal · Authors · 2024-08-06
>
> We sincerely thank Reviewer RKXd for the thoughtful and constructive reviews of our manuscript! Based on your questions and recommendations, we give point-by-point responses to your comments and describe the revisions we made to address them.
>
> ---
> > **Weakness 1**: Though the authors claim that GDER can be extended to graph regression, more experiments are needed to validate their claim in Section 4.2.
>
> Thank you for your insightful comment! We would like to respectfully point out that in Section 3.5 and Appendix D, we provide a detailed explanation of how to extend GDeR to the scenarios of graph regression or unsupervised learning. In Observation 3 of Section 4.2, we analyze the pruning performance and efficiency acceleration of GDeR on GraphMAE+ZINC. We hope this can help you better understand the versatility of GDeR.
>
> ---
> > **Weakness 2**:  In robust GNN training, the baselines compared are outdated. There are stronger robust GNN baselines [1,2,3].
>
> We admire your extensive knowledge! As the methods you cited [1,2] are limited to node classification, we supplemented the comparison with GraphDE [3] and the latest SOTA method MRGL [4] with our GDeR. Additionally, we employed a newer graph attack method GRABNEL [5] for perturbation attacks. As shown in Table A, GDeR outperforms GraphDE and MRGL in high noise scenarios through its unique soft pruning mechanism.
>
> *Table A. F1-Macro (%) comparison among GraphDE, MRGL and GDeR on DAGCN+MUTAG with GRABNEL. We set the pruning ratio of GDeR to 30%.*
> |Noise ratio|0%|5%|10%|20%|
> |-|-|-|-|-|
> |GraphDE|85.12|82.99|81.45|79.16|
> |MRGL|85.12|**84.66**|82.07|78.59|
> |GDeR|85.12|83.08|**83.46**|**80.14**|
>
> [3] Graphde: A generative framework for debiased learning and out-of-distribution detection on graphs, NeurIPS 2022
>
> [4] Multi-View Robust Graph Representation Learning for Graph Classification, IJCAI 2023
>
> [5] Adversarial Attacks on Graph Classification via Bayesian Optimisation， NeurIPS 2021
>
> ---
> > **Weakness 3**: The authors merely test GDER with poisoning attack on node features. Is GDER capable of addresing evasion attack? Also, the experiments can be stronger if the authors use more advanced GNN attack methods like Mettack [4] or Nettack [5].
>
> Thank you for your feedback! Currently, GDeR only employs defense against adversarial poisoning attacks as it can dynamically identify and remove noisy samples during training. However, we would like to respectfully state this does not affect the comtribution of our work as we understand that most current robust GNN methods focus on solving poisoning attacks rather than evasion attacks.
>
> In our manuscript, we only used noise perturbation on node features as the attack method. This is because there are currently few attack methods specifically designed for graph-level classification tasks, as noted in [1]. However, we have supplemented in our response to Weakness 2 with a performance comparison of different robust GNN methods against the newer graph-level attack GRABNEL, and we hope this will address your concerns.
>
> [1] Adversarial Attacks on Graph Classification via Bayesian Optimisation
>
> ---
> > **Minor**
>
> Thank you for your insightful suggestions! We promise to include Table 7 in the main text in the camera-ready version. Regarding Line 184/189, we did mistakenly write $\mathcal{X}_t$ as $\mathcal{D}^{(t)}$, and we appreciate your correction!
>
> ---
> > **Question 1**: Figure 1 shows the sample label ditrbution using Infobatch, right? If so, what is the distribution like with GDER?
>
> Thank you for making our work even stronger! We tracked the training trajectory and sample distribution of GDeR on MUTAG+GCN. **The visualizations are available at the Figure 1 of the global rebuttal PDF**. As can be seen, unlike InfoBatch, which amplifies sample imbalance, our GDeR can effectively alleviate the problem of imbalance by allocating more samples from minority classes and achieving a stable distribution in the later stages of training.
>
> ---
> > **Question 2**: Can you give a case study of the graph samples pruned by GDER? For examples, how many of them are with noise? how many of them are from the majority class?
>
> Thank you for your comment, which has further improved the quality of our article!
>
> - Regarding **pruning for noisy data**, we tracked the training trajectory of GDeR and visualized how it balances the distribution of noisy and clean data in noisy scenarios. **The visualizations are available at the Figure 2 of the global rebuttal PDF**. It can be observed that as training progresses, GDeR gradually corrects the distribution of the training set, reducing the proportion of noisy/biased data in the maintained training set and thereby reducing the impact of biased samples on the model.
> - Regarding **pruning of imbalanced data**, please refer to our response to Question 1.

---

> > ### Comment · Reviewer_RKXd · 2024-08-10
> > **Response**
> >
> > I read the replies, which addressed most of my concerns. I would like to raise my score.

---

> > > ### Author Response · Authors · 2024-08-10
> > >
> > > We are glad that our rebuttal has effectively addressed you concerns! We are still open to any further questions you may have :)

---

### Author Rebuttal · Authors · 2024-08-06

Dear Reviewers,

We extend our sincere gratitude for your dedication to the review process. We are truly encouraged by the reviewers' recognition of several positive aspects of our paper, including **a novel and significant data pruning method** (Reviewer `RKXd`,  `kNzQ`, `zhEK`), **high-quality presentation** (Reviewers  `RKXd`,  `kNzQ`, `zhEK`), and **encouraging experimental rsults** (Reviewers `RKXd`). Additionally, we have diligently addressed and responded to each concern raised by every reviewer. Here, we summarize several key points as follows:

1. **Transferability to other data domains** (Reviewer `kNzQ`, `zhEK`)
To demonstrate the transferability of GDeR to other data domains, we present its performance on CIFAR-10 and ImageNet datasets (in Tables 1 and 2 of the gloabl rebuttal PDF).

2. **Updated baselines** (Reviewer `RKXd`, `zhEK`)
We compare GDeR with more advanced robust GNN and long-tail techniques (in Tables 3 and 4).

3. **Complexity and efficiency** (Reviewer `zhEK`)
We supplement the complexity analysis of GDeR and provide corresponding wall-clock time evaluation (in Tables 5 and 6).

4. **Case Study** (Reviewer `RKXd`)
We have supplemented a case study, demonstrating how GDeR dynamically removes noisy data and alleviates long-tail distribution in practical training processes (in Figures 1 and 2).

Thanks again to all the reviewers, and your expertise significantly helps us strengthen our manuscript – this might be the most helpful review we received in years! We are willing to address any further concerns or questions you may have.

Warm regards,

Authors

---

### Decision · Program_Chairs · 2024-09-25

**Decision:**

Accept (poster)

**Comment:**

Despite the fact that there has been a good number of data pruning techniques aiming at saving the neural network training computation costs, it seems that no one has explored the application of data pruning techniques for graph-level data in graph neural network training.

This paper, for the first time, systematically evaluated quite a few data pruning techniques that have been previously used for computer vision or NLP tasks on pruning graph-level datasets, specifically focusing on accelerating neural network training, mitigating the dataset (class) imbalance issue, and improving neural network robustness in the presence of adversarial attacks. The paper found that all the existing methods being tested in the paper display issues in training efficiency, imbalanced learning, and robustness learning. Based on these findings, the paper proposed a new data pruning method "Graph De-Redundancy" (GDeR) and demonstrated superior performance in all the three aspects than the other baseline methods. Specifically, GDeR adds additional modules to GNN to project the intermediate data sample feature representation vector to hyperspherical embedding space with learnable center prototype representations for each class in the dataset and encourages intra-class compactness and inter-class separations. Using the learned hyperspherical embeddings, a data sampling method that considers sample anomaly, familiarity, and balance was proposed for soft-pruning (sampling) dataset to accelerate training. The GDeR method has set up a good baseline for future data pruning techniques that primarily focus on graph-level data pruning.

While the paper has conducted systematic evaluations and set up a good baseline for this specific direction for the first time, there are a few items that might worth taking into consideration for further improvements.

1. There are typos or unannotated symbols in the equations. Even if the reader might be able to guess what they are, we have to ensure the correctness and clarity of the paper anyway. For example, what is $k$ and $\mathcal{N}$ in the Equation 1? What is $I$ and $d$ in the Equation 3? Please proof reading all the mathematical equations and their annotations in the paper, ensuring there is no error.
2. The novelty of the proposed GDeR method has not been described in detail in the paper. GDeR works on feature embeddings after all. It has been inspired by many other previous works on representation learning and data coreset selection. The paper should provide references to the equations related to the previous work. For example, even though I cannot fully remember where the source is, I am pretty sure I have seen previous representation learning works that encourage intra-class compactness and inter-class separations for learned embeddings of different classes. Of course, the paper should also highlight the key technical distinctions and innovations in GDeR comparing to the previous works.
3. The paper has done a tremendous amount of work evaluating the previous data pruning methods on graph-level data pruning, for the first time. These work should be mentioned and emphasized in the key locations of the paper, including the abstract, to motivate the proposal of the new GDeR method.
4. Need to clarify or fix "there is currently no method for graph-level data pruning" in line 119. This is a very strong statement and shall not confuse the readers. All the methods presented in this paper can be used for graph-level data pruning, it's just never been systematically tested for it.